# Substrate binding-induced conformational transitions in the omega-3 fatty acid transporter MFSD2A

Shana Bergman[1,4], Rosemary J. Cater[2,4], Ambrose Plante[1], Filippo Mancia[2] & George Khelashvili [1,3] ✉

Major Facilitator Superfamily Domain containing 2 A (MFSD2A) is a transporter that is highly enriched at the blood-brain and blood-retinal barriers, where it mediates Na$^+$-dependent uptake of ω−3 fatty acids in the form of lysolipids into the brain and eyes, respectively. Despite recent structural insights, it remains unclear how this process is initiated, and driven by Na$^+$. Here, we perform Molecular Dynamics simulations which demonstrate that substrates enter outward facing MFSD2A from the outer leaflet of the membrane via lateral openings between transmembrane helices 5/8 and 2/11. The substrate headgroup enters first and engages in Na$^+$-bridged interactions with a conserved glutamic acid, while the tail is surrounded by hydrophobic residues. This binding mode is consistent with a "trap-and-flip" mechanism and triggers transition to an occluded conformation. Furthermore, using machine learning analysis, we identify key elements that enable these transitions. These results advance our molecular understanding of the MFSD2A transport cycle.

Endothelial cells of the blood-brain and blood-retinal barriers are highly enriched in Major Facilitator Superfamily Domain containing 2A (MFSD2A)[1]. MFSD2A is an integral membrane transporter that mediates Na$^+$-dependent uptake of ω−3 fatty acids like docosahexaenoic acid (DHA) in the form of zwitterionic lysolipids such as lysophosphatidylcholine (LPC-DHA) into the brain and eyes, respectively[2–5]. In humans, single nucleotide polymorphisms in *MFSD2A* and changes in MFSD2A expression levels have been implicated in several severe neurological disorders including autosomal recessive primary microcephaly[6–11], intracranial haemorrhage and Alzheimer's disease[12].

MFSD2A belongs to the Major Facilitator Superfamily (MFS) of transporters[13]. Like most other members of the superfamily, this transporter is comprised of 12 transmembrane helices (TMs) organised into two pseudosymmetric six-helix bundles called the N-terminal domain (TMs 1-6) and the C-terminal domain (TMs 7-12)[14]. The majority of MFS transporters mediate the import/export of water-soluble molecules using a "rocker-switch" mechanism[13]. According to this mechanism, the N- and C-terminal domains undergo rigid-body movements around a centrally located substrate-binding site, alternatively exposing it to the extracellular (EC) and intracellular (IC) sides of the membrane. These outward-facing and inward-facing states (OFS and IFS, respectively) are connected in the context of the transport cycle through an occluded state (OcS) in which the substrate binding site is inaccessible from either side of the membrane[15].

Recently, we determined the structure of MFSD2A from *Gallus gallus* (ggMFSD2A) in an IFS using single-particle cryo-electron microscopy (cryo-EM; Fig. 1a)[14]. In this conformation, two major functional sites of the transporter, a pair of absolutely conserved and functionally essential residues (E312 and R85) and the proposed Na$^+$-binding site[2] (D92, Fig. 1a, b), are located in a large IC-accessible cavity, in the central region of the protein. This cavity also harbours the substrate LPC-18:3 in a head-down configuration with its acyl tail buried in a hydrophobic pocket that spans about two thirds of the membrane height (Fig. 1a, b). By combining these structural insights with functional analyses and large-scale molecular dynamics (MD) simulations, we discovered that binding of Na$^+$ at D92 facilitates

[1]Department of Physiology and Biophysics, Weill Cornell Medical College, Cornell University, New York, NY 10065, USA. [2]Department of Physiology and Cellular Biophysics, Columbia University, New York, NY 10032, USA. [3]Institute for Computational Biomedicine, Weill Cornell Medical College, Cornell University, New York, NY 10065, USA. [4]These authors contributed equally: Shana Bergman, Rosemary J. Cater. ✉e-mail: gek2009@med.cornell.edu

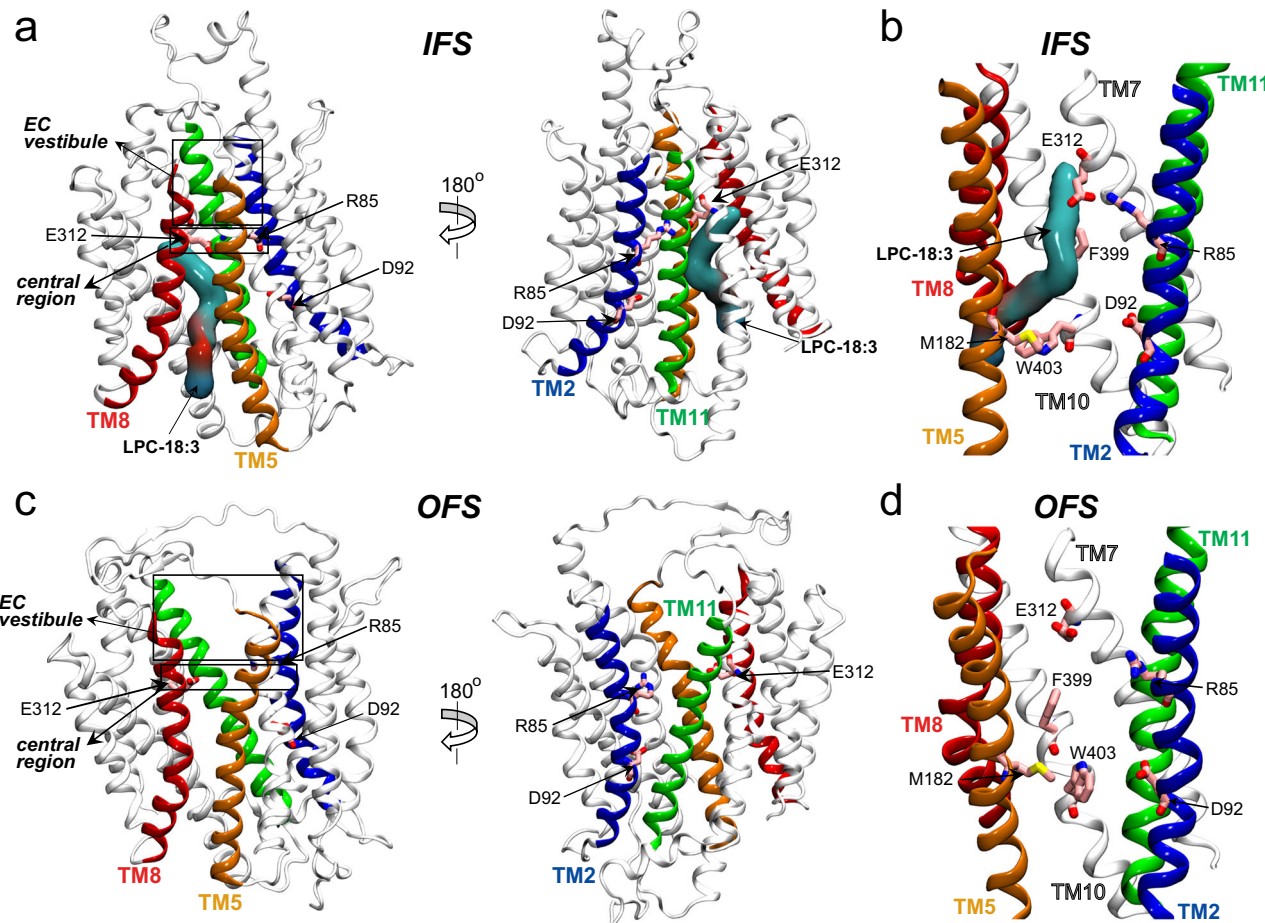

**Fig. 1 | The IFS and OFS of MFSD2A. a** Two views of ggMFSD2A in the IFS (PDBID: 7MJS[14]) from the plane of the membrane, related by a 180° rotation. TMs 2, 5, 8 and 11 are shown in blue, orange, red and green, respectively. The remainder of the protein is coloured in light grey. The substrate, LPC-18:3, is shown in surface representation. **b** Detailed view of the central region in the IFS highlighting (as pink sticks) the central charged residues E312 and R85, the Na$^+$ binding site residue D92, and the IC gate residues, M182, F399 and W403. All TMs except TMs 2, 5, 7, 8, 10 and 11 are omitted for visual clarity. **c, d** The equivalent of panels **a** and **b**, respectively, for the homology model of ggMFSD2A in the OFS. The locations of the EC vestibule and central region are highlighted in panels **a** and **c** with rectangular boxes.

opening of an IC gate between TMs 5, 8 and 10 (comprised of residues M182, F399 and W403; Fig. 1b), through which lysolipid substrates can be released directly into the inner leaflet of the membrane. Our MD simulations also demonstrated that within the central region, the substrate headgroup can be coordinated by E312 and R85, and that Na$^+$ can transiently interact with E312 in the absence of substrate[14].

Parallel and subsequent studies have reported the structures of MFSD2A in other conformations[16,17], including one of *Mus musculus* MFSD2A (mmMFSD2A) in an OFS[16] (Fig. 1c-d). In this structure, E312 and R85 are exposed to the EC milieu via a vestibule lined by the EC ends of TMs 1, 2, 5, 7, 8 and 11 (EC vestibule; Fig. 1c). Comparison of the OFS and IFS reveals large-scale conformational changes of two helical pairs, TM5/TM8 and TM2/TM11, each containing one helix per domain. Indeed, in the OFS these two pairs create a right-side-up V-shape (Fig. 1c), whereas in the IFS, they are reoriented to adopt an inverted V-shape (Fig. 1a). Interestingly, the OFS structure featured lipid-like cryo-EM densities within lateral openings at the membrane-protein interface, between TM5/TM8 and TM2/TM11, suggesting that they may provide an entrance pathway for substrates from the outer leaflet of the membrane into the EC vestibule[16].

Together, these studies demonstrate that MFSD2A utilises a "rocker-switch" mechanism of alternating access, but it is not known how MFSD2A has adapted its mode of substrate engagement to allow for the transport of lysolipids – which are atypical substrates for MFS transporters. Broadly speaking, two different models of protein-

mediated lipid transport have been proposed, the "trap-and-flip", and the "credit-card" mechanism[18–27]. In the "trap-and flip" model, lipid substrates are recruited via the membrane and entirely enclosed by the transporter as they are shuttled from one leaflet of the membrane to the other. This was recently demonstrated to be the mechanism by which the proton-dependent MFS glycolipid exporter LtaA functions[27]. In contrast, in the "credit-card" model the lipid headgroups traverse the membrane by populating a hydrophilic pathway in the transporter, while the hydrophobic tails are maintained in the hydrophobic milieu of the lipid bilayer. Lipid scramblases such as those from the TMEM16 family have been shown to utilise this mechanism[28]. Importantly, in the context of MFSD2A, both of these mechanisms would be adaptations of the "rocker-switch" mechanism – with the "trap and flip" or "credit card" component describing how the lysolipid substrate engages with the protein during the translocation process, and the "rocker-switch" describing the conformational changes the protein undergoes to accommodate the substrate and facilitate its transport.

In this study, we focus on several key mechanistic questions to delineate if MFSD2A utilises a "trap and flip" or "credit card" adaptation of the "rocker-switch" mechanism to allow for transport of lysolipid substrates: (i) How do substrates enter and bind to MFSD2A on the extracellular side? (ii) What is the role of Na$^+$ in this process? and (iii) How does substrate binding facilitate isomerization to an OcS? To answer these questions, we perform extensive atomistic multi-replicate MD simulations of MFSD2A which reveal that substrate

enters MFSD2A from the outer leaflet of the lipid bilayer via both the TM5/TM8 and TM2/TM11 lateral openings. Once within the central region, the headgroup of the substrate binds to E312 in a Na$^+$-bridged manner and the tail is engulfed by hydrophobic residues on the EC side. Our machine learning-based analysis of these simulations further demonstrate how this substrate binding mode is allosterically coupled to conformational changes on the IC side of the transporter that allow for transition of MFSD2A to the OcS and then IFS. These findings complement available structural and functional studies, provide a comprehensive molecular-level understanding of the transport cycle, and support the notion that MFSD2A utilises a "trap-and-flip" adapted "rocker-switch" mechanism.

## Results

### Substrates enter MFSD2A via lateral openings to the membrane
To understand how the substrate enters and binds to MFSD2A, we carried out 48 independent atomistic multi-replicate MD simulations of an outward-facing ggMFSD2A model (Fig. 1c, constructed based on homology to mmMFSD2A, see Methods) in a phosphatidylcholine (POPC) bilayer containing the substrate LPC-18:1 in the EC leaflet. During these simulations, which had a cumulative sampling time of ~79 µs, we observed multiple events of LPC-18:1 insertion into the central region of the protein via the lateral openings between both TM5/TM8 and TM2/TM11 (Fig. 2a, b). To quantify these insertion events, we defined the C$_\alpha$ atom of F399 (part of the IC gate located on TM10, Figs. 1c, d and 2a) as a reference point, and measured the minimal distance along the membrane normal (z) axis ($d_Z$), between every LPC-18:1 lipid phosphorus atom (P) and F399$_{C\alpha}$ in each trajectory frame. This analysis revealed three distinct populations of LPC-18:1 (Fig. 2c): one with the LPC-18:1 headgroup positioned in the bulk membrane ($d_Z$ ~ 20 Å); a second with the LPC-18:1 headgroup positioned near E312 in the central region ($d_Z$ ~ 13 Å); and a third with the LPC-18:1 headgroup further down towards the IC side, and in close proximity to the IC gate ($d_Z$ ~ 5 Å, see also Fig. 2a). On the simulation timescales, the number of LPC-18:1 molecules that inserted themselves into the central region (i.e., those with $d_Z \leq 13$ Å) via each lateral entrance varied. From our 48 trajectories, we observed that LPC-18:1 entered via TM5/TM8 10 times and via TM2/TM11 21 times. Interestingly, there were also 4 instances in which the central region was penetrated by the substrates from both sides concomitantly.

In contrast to what we observed with LPC-18:1, our equally extensive control simulations of ggMFSD2A embedded in a pure POPC membrane (48 replicates, ~74 µs cumulative sampling time) revealed that POPC molecules are unable to insert themselves deep into the transporter (i.e., near the IC gate). Indeed, while POPC lipid headgroup penetration was observed in 8 simulations – exclusively via the TM2/TM11 entrance – the headgroup did not reach the level of F399$_{C\alpha}$ (Fig. 2d–f). This behaviour is in line with observations from our previous MD simulations of MFSD2A in the IFS, wherein POPC could only partially and transiently enter the intracellular cavity, but lysolipid substrates (LPC-18:1, LPC-18:3, LPC-DHA) could extensively sample and penetrate deeply into it[14]. These observations can likely be attributed to the relatively larger size of the two-tailed POPC compared to the single-tailed LPC-18:1 (Fig. 2g), with estimated volumes of ~1265 Å$^3$ and ~822 Å$^3$ at 40 °C, respectively[29]. Overall, these results demonstrate that lysolipids are more likely to insert with their headgroup deep into the transporter than glycerophospholipids, consistent with the experimental observation that MFSD2A specifically transports mono-acyl chain lipids[2].

### Na$^+$ mediates interactions between the substrate headgroup and E312
While most substrate insertion events were partial – i.e., only the headgroup would insert into the central region while the lipid tail would remain in the outer leaflet of the membrane – we observed two

events (trajectories) of substrate insertion via the TM5/TM8 pathway that resulted in full substrate embedding and occlusion in the central region. In the first trajectory (Traj-1), only one substrate was seen penetrating (via the TM5/TM8 lateral entrance) and inserting into the central region (Supplementary Movie 1). In the second trajectory (Traj-2), substrate penetration via the TM5/TM8 lateral entrance occurred concomitantly with a second substrate entering via the TM2/TM11 lateral entrance (Supplementary Movie 2). The binding modes of the substrates that inserted via the TM5/TM8 entrance in these two trajectories were similar, with the substrate headgroup coming in close proximity to E312 (Fig. 3a, b). In Traj-2, the headgroup of the additional substrate that entered via TM2/TM11 formed contacts with the fully inserted substrate (that entered via TM5/TM8), while its tail predominantly remained in the lipid bilayer.

In both trajectories, the interaction between the substrate headgroup and the carboxyl group of E312 were long-lasting (up to 800–900 ns timescale), and – surprisingly – bridged by Na$^+$ (Fig. 3c). Indeed, when Na$^+$ was within 2.5 Å of E312, the P atom of the substrate was stably maintained at <6 Å from the E312 carboxyl carbon atom (Fig. 3b, c). In this configuration, the Na$^+$ bridges the carboxyl moiety of E312 and the oxygen atoms of the LPC headgroup (Fig. 3d–f). This is different from what we observed in the simulations without substrates as well as in our previous simulations of substrate-free ggMFSD2A in the IFS[14], where Na$^+$ only transiently interacted with E312 (on the timescales of <100 ns) and the LPC headgroup could directly, albeit transiently, engage with E312. Together, these results suggest binding of Na$^+$ and of the substrate headgroup at E312 are synergistic and provide mutual stabilisation at this site.

Interestingly, coordinated binding of the substrate headgroup and Na$^+$ by E312 was accompanied by conformational shifts in E312 and in the nearby residue F399 which constitutes part of the IC gate. Specifically, during Traj-1, the sidechain of E312 gradually rotated from facing TM10 to facing TM2, whereas the aromatic ring of F399 switched from facing TM7 to TM5 (Fig. 3g). These dynamic changes may be required to accommodate the substrate headgroup in the central region.

### E312D substitution destabilises LPC-18:1 in the binding site
We previously demonstrated that the conservative mutation E312D is sufficient to significantly abrogate MFSD2A-mediated transport[14]. This, combined with our observation that Na$^+$ mediates the interaction between the substrate headgroup and E312 (Fig. 3), allows us to hypothesise that the detrimental effect of this mutation could be due to disruption of the E312-Na$^+$-substrate interaction. To test this, we introduced E312D into two different frames from the Traj-1 simulations – differing only in the rotameric state of F399 to explore if the conformation of this IC gate residue influences substrate binding in parallel – and carried out ~36 µs long multi-replicate simulations, alongside an ~50 µs long multi-replicate simulation from the same frames in the wild type (WT) system (Fig. 3d, e).

To quantify the differences between these simulations, we calculated the root-mean-square deviation (RMSD) of the substrate position and measured the distance between its P atom and the carboxyl carbon of the 312 sidechain (Fig. 4a, b). For E312D, this analysis showed diffuse RMSD profiles for substrate position and broad distance distributions, suggesting destabilization. In contrast, the RMSD of the substrate position in the WT system had a distinguished peak at low values, and the P-E312 distance distribution was narrow, with a peak at ~5 Å. These data are consistent with the substrate being more stably bound in the WT protein.

These effects appear to be related to the different modes of Na$^+$ binding at position 312 in the two systems. Indeed, in the WT system, the distance between the carboxyl carbon of E312 and the nearest Na$^+$ ion is ~2.5 Å (Fig. 4c, Peak P1), corresponding to Na$^+$ coordination that involves both carboxyl oxygens of the E312 sidechain (Fig. 4d). In

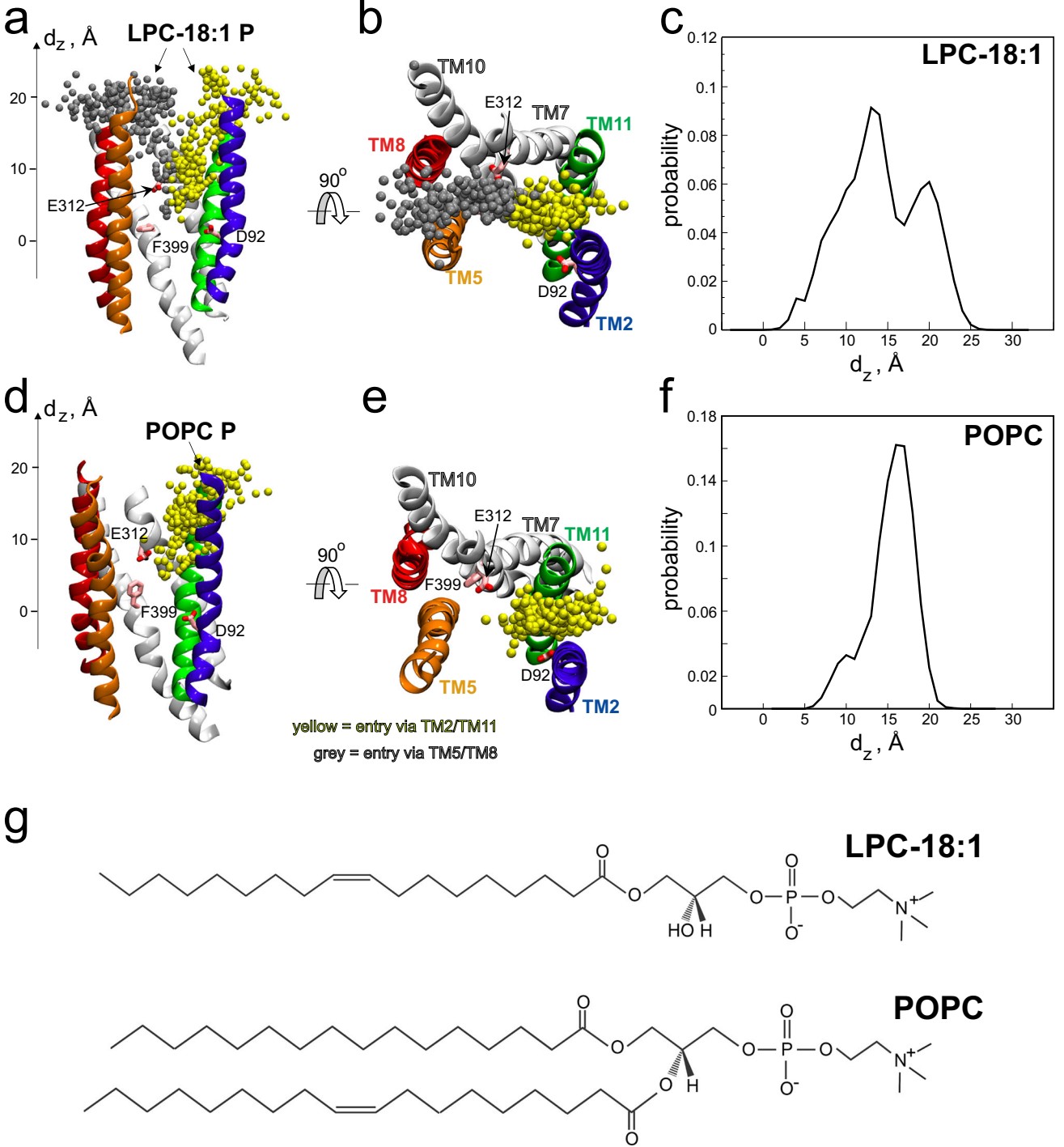

**Fig. 2 | Pathways for LPC-18:1 entry into the MFSD2A central region.**
**a** Superposition of LPC-18:1 phosphorus (P) atom in the MD trajectory frames shown as spheres. Only positions within 5 Å of the protein are included, and the atoms penetrating the central region from the two different pathways, TM5/TM8 and TM2/TM11 are coloured in grey and yellow, respectively. The protein is otherwise represented as in Fig. 1. **b** The same as in panel **a** rotated by 90° to provide the view from the EC side. **c** Histogram of the minimum z-directional distance between LPC-18:1 P and the $C_\alpha$ of F399 ($d_z$, $z = 0$ represents z-position of the $C_\alpha$ atom of F399). **d**–**f** The same representations as in panels a-c only for the P of POPC from the MD simulations of ggMFSD2A in a pure POPC membrane. **g** Comparison of the chemical structures of LPC-18:1 and POPC.

contrast, for E312D, peak P1 is diminished, and a second peak P2 is formed at >3 Å separation, corresponding to a binding mode in which Na⁺ is coordinated by just one carboxyl oxygen of D312 (Fig. 4e). Taken together, these data combined with our prior functional characterisation of E312D[14], are consistent with the requirement of Na⁺ for optimal coordination and stabilisation of the substrate headgroup via both terminal oxygens of E312.

**The substrate hydrophobic tail is stabilised by aromatic residues in the EC vestibule**

While the substrate headgroup interacts with E312 as described above (Fig. 4), its hydrophobic tail is engulfed within the transporter and adopts multiple positions as it explores the EC vestibule (Fig. 5a and Supplementary Movies 1 and 2). In Traj-1, following initial substrate insertion (Fig. 5a; segment I), the terminal end of the tail is situated

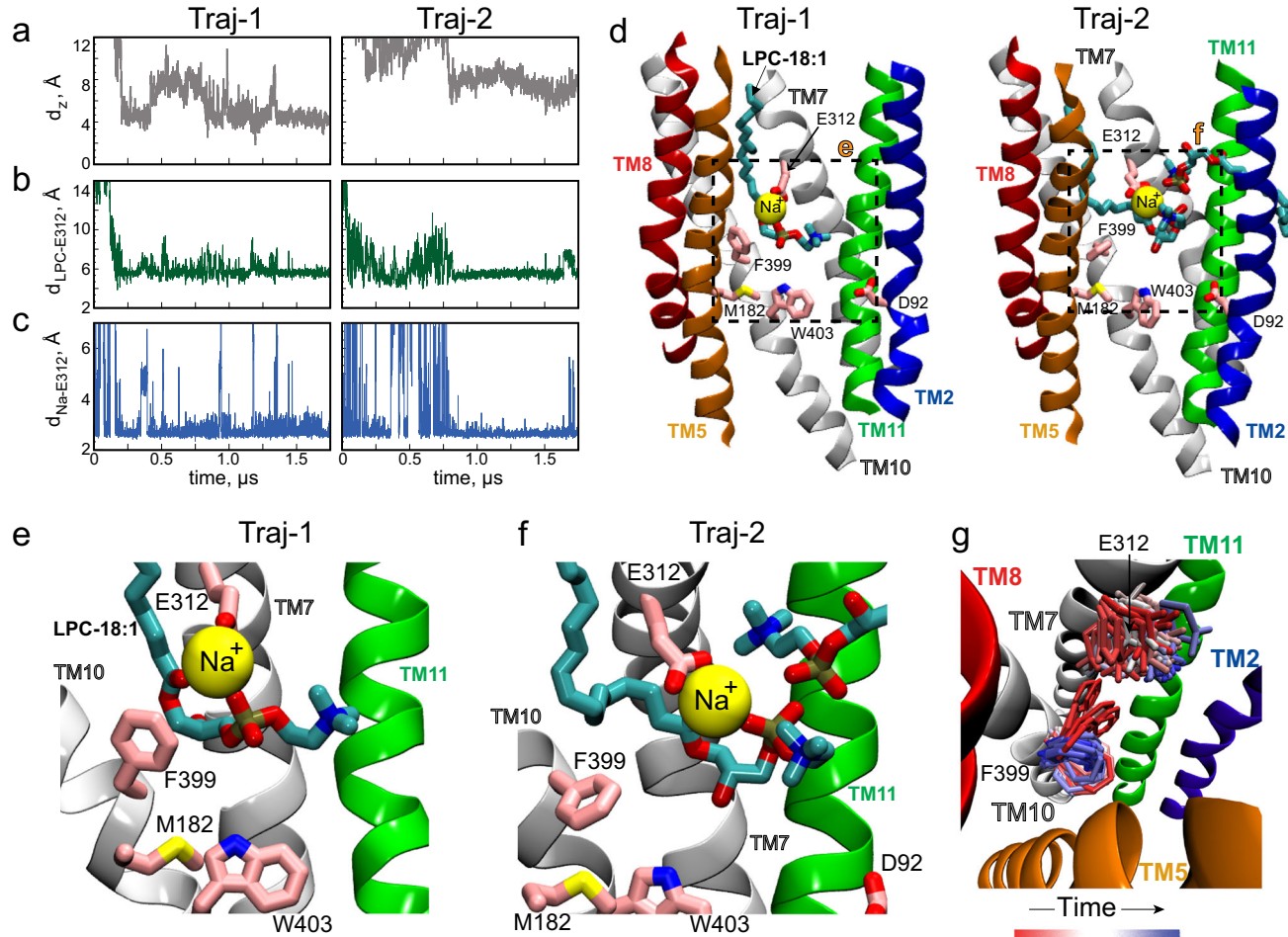

**Fig. 3 | LPC-18:1 binds stably within the central region of MFSD2A.** Time-evolution of **a** the distance along the vertical *z*-axis between the phosphorus (P) atom of the penetrating LPC-18:1 and the $C_\alpha$ of F399 ($d_z$), **b** the distance between the carboxyl carbon atom of E312 and the P of the penetrating LPC-18:1, and **c** the distance between the carboxyl carbon atom of E312 and the closest $Na^+$ in two separate MD trajectories (Traj-1 and Traj-2). **d** Representative snapshots from Traj-1 (left) and Traj-2 (right) showing LPC-18:1 insertion. LPC-18:1 is shown in stick representation and the protein is represented as in previous figures. **e, f** Detailed view of the protein central region from the snapshots shown in panel **d**, highlighting the modes of LPC-18:1 binding in Traj-1 and Traj-2. **g** Conformations of E312 and F399 from the trajectory frames of Traj-1 superimposed on the initial structure of the protein. The sidechain positions are coloured according to time, with red representing the starting frames, white – the middle, and blue – the last third of the trajectory.

close to the EC end of TM1, whereas the region proximal to the headgroup is located near TM10. Indeed, both the distance between the LPC-18:1 ω−1 carbon (in IUPAC numbering) and F60, located towards the top of TM1, and the distance between the LPC-18:1 ω−14 carbon and V395, located approximately halfway along TM10, are ~5 Å (Fig. 5a; segment I). After ~0.4 µs the substrate tail rearranges so that the ω−1 carbon moves away from TM1 and towards TM8 (Fig. 5a; segment II). As a result, the ω−1/F60 distance increases to ~14 Å and conversely, the distance between the ω−1 carbon and L333 in TM8 decreases from ~10 Å in segment I to ~5 Å in segment II (Fig. 5a). After ~1 µs, the substrate tail repositions again so that its terminal end moves back towards TM1, resulting in an ω−1/F60 distance of ~4 Å. In concert with this, the headgroup-proximal region of the tail becomes sandwiched between TM7 and TM10, as indicated by an ω−14/V395 distance of ~4 Å where it is then maintained until the end of the simulation (Fig. 5a, segment III).

In Traj-2 – where the central region is penetrated by two substrates via the TM5/TM8 and TM2/TM11 lateral entrances concomitantly (Fig. 3d) – the substrate tail inserting from the TM5/TM8 entrance adopts a position similar to that observed in Traj-1 segment II (Fig. 5b, segment IV). The terminal region of the tail then samples an area between TM7 and TM10, as indicated by an increased ω−1/L333

distance (Fig. 5b, segment V). In contrast, the ω−14 carbon region is less dynamic, maintaining a distance of ~5 Å to V395 throughout segments IV and V. The somewhat different positioning of the substrate tail in Traj-2 compared to that of Traj-1 may be attributed to the fully embedded LPC-18:1 in the former also being engaged with the headgroup of the other partially inserted substrate (Fig. 3d and Supplementary Movie 2). Overall, this analysis shows that following an initial dynamic sampling of the EC vestibule, the terminal region of the hydrocarbon tail of the inserted substrate is stabilized by a network of aromatic residues, whereas the headgroup-proximal region is stably positioned between TM7 and TM10.

## LPC-18:1 insertion leads to dehydration of the EC vestibule and transition to an OcS

Next, we investigated the conformational changes in MFSD2A that accompany substrate insertion. We reasoned that the EC vestibule would narrow upon substrate insertion to allow its hydrophobic residues to coordinate the inserted substrate tail as described above (Fig. 5). To probe this hypothesis, we measured the distances between the $C_\alpha$ atoms of residues on the EC ends of TM1 (F61), TM5 (V200), TM7 (A316) and TM8 (F329) throughout the duration of Traj-1 and Traj-2. Over the course of Traj-1, these distances contract on average by ~2 Å

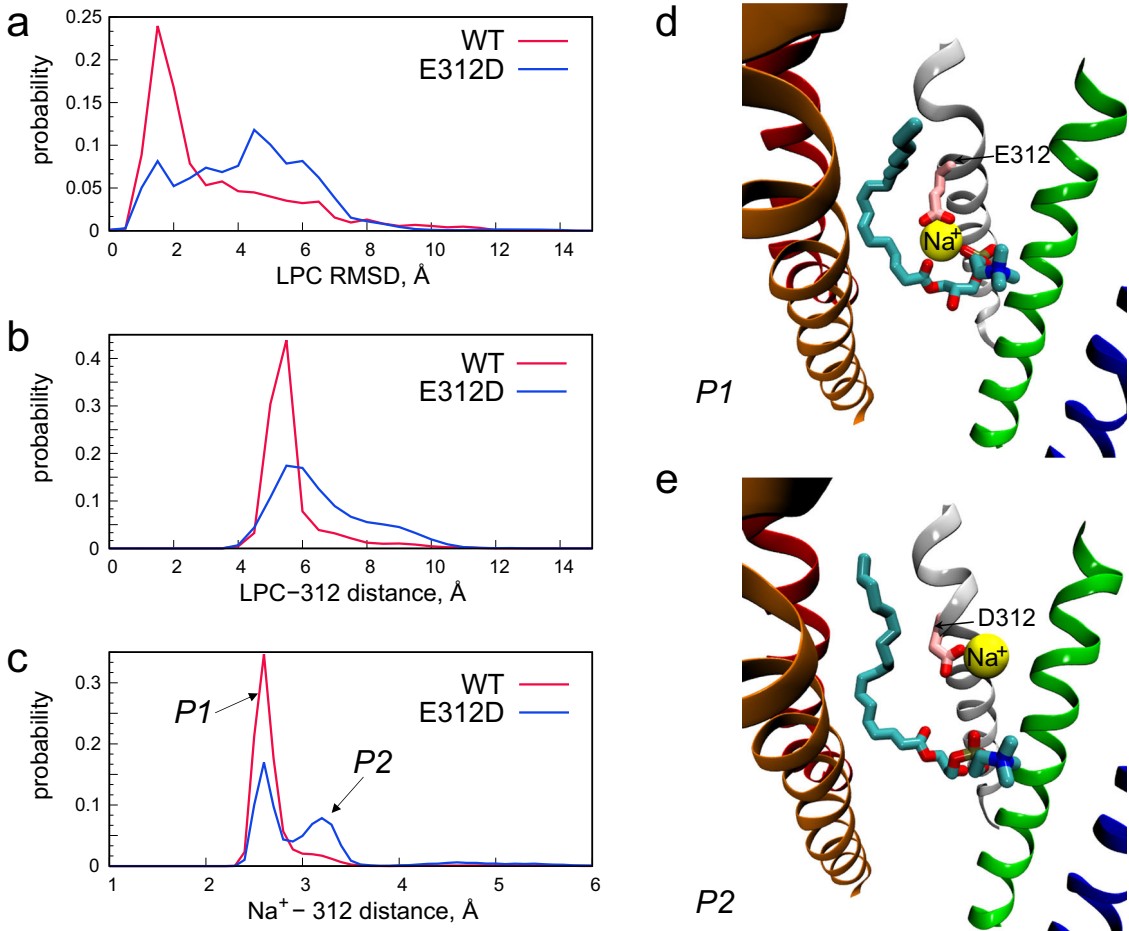

**Fig. 4 | The E312D mutation destabilises LPC-18:1 binding. a** Histogram of LPC-18:1 RMSD in the WT and E312D systems. The RMSD of non-hydrogen LPC-18:1 atoms was calculated after aligning the trajectories using the initial coordinates of the backbone atoms of the protein TMs as reference. Histograms of the distances between the carboxyl carbon of residue 312 in the WT and E312D systems and **b** the LPC-18:1 P and **c** the nearest $Na^+$ ion. The two peaks P1 and P2 are labelled.

Snapshots of **d** WT ggMFSD2A in the P1 conformation and **e** E312D ggMFSD2A in the P2 conformation illustrating the different modes of $Na^+$ binding. LPC-18:1 is shown in stick representation, and $Na^+$ as a yellow sphere. The protein is otherwise represented as in previous figures with all TMs except TMs 2, 5, 7, 8 and 11 omitted for visual clarity.

(Fig. 6a), resulting in an OcS conformation in which both the TM5/TM8 lateral entrance and the EC vestibule have substantially narrowed, and the substrate is entirely surrounded by protein (Fig. 6c). Consistent with these changes, the number of accumulated water molecules following substrate insertion is also reduced from ~45 to ~30 (Fig. 6b), suggesting that this insertion results in partial dehydration of the EC vestibule. In contrast, the extent of EC vestibule closure and dehydration in Traj-2 is not as pronounced (Fig. 6a, b). This is not surprising given the simultaneous presence of the second lysolipid, which has its hydrocarbon tail extending through TM2/TM11 and into the membrane, likely impacting the protein dynamics that enable occlusion of the transporter. Interestingly, we found that closure of the TM5/TM8 lateral entrance upon occlusion was largely induced by a kinking motion around the conserved residue P345 on the IC end of TM8 (Fig. 6d). Indeed, the distribution of the kink angle for the two trajectories in which substrate insertion and concomitant formation of the OcS occurred was reduced (~34° on average) compared to simulations which led to no lipid penetration or occlusion (~40° on average; Fig. 6e). It is possible that this kinking may be mechanistically important for regulating the equilibrium between the open/closed state of the intracellular TM5/TM8 lateral gateway through which lysolipids exit the central cavity[14].

To better understand how these helical movements that occur during occlusion affect the overall distribution of waters inside the

protein, we carried out a pore analysis of both the OFS and OcS (Fig. 7). These analyses revealed that MFSD2A contains a continuous water pore running through its interior in both states, and that the 'central region' of this channel (demarcated by residues F399 and F60; pore coordinate 135–150) is narrower in the OcS than in the OFS (Fig. 7a–c). Interestingly, the direction of the pore axis in this region diverges in the OFS and OcS trajectory sets. Indeed, in the OFS, the pore axis is relatively central, whereas in the OcS it is diverted towards the TM2/TM11 region (Fig. 7d–f). This is likely due to the inward motion of TMs that line the EC vestibule during occlusion (TM 1, 5, 7 and 8; Fig. 6) which narrows the EC vestibule, consequently redirecting the water pathway towards TM2/TM11.

## TM7 is a major structural element for classification of the OFS and OcS conformations
Next, we investigated specific conformational changes that initiate isomerization of MFSD2A into the OcS described above (Fig. 6). To achieve this, we discerned structural differences between the OFS and OcS using a Deep Neural Network (DNN) classification algorithm for pattern recognition analysis of MD trajectories that we previously developed[30]. This machine learning protocol transforms function-related, construct/state-specific differences encoded in MD trajectories into visual representations recognisable by deep learning object recognition technology. The method then performs classification tasks

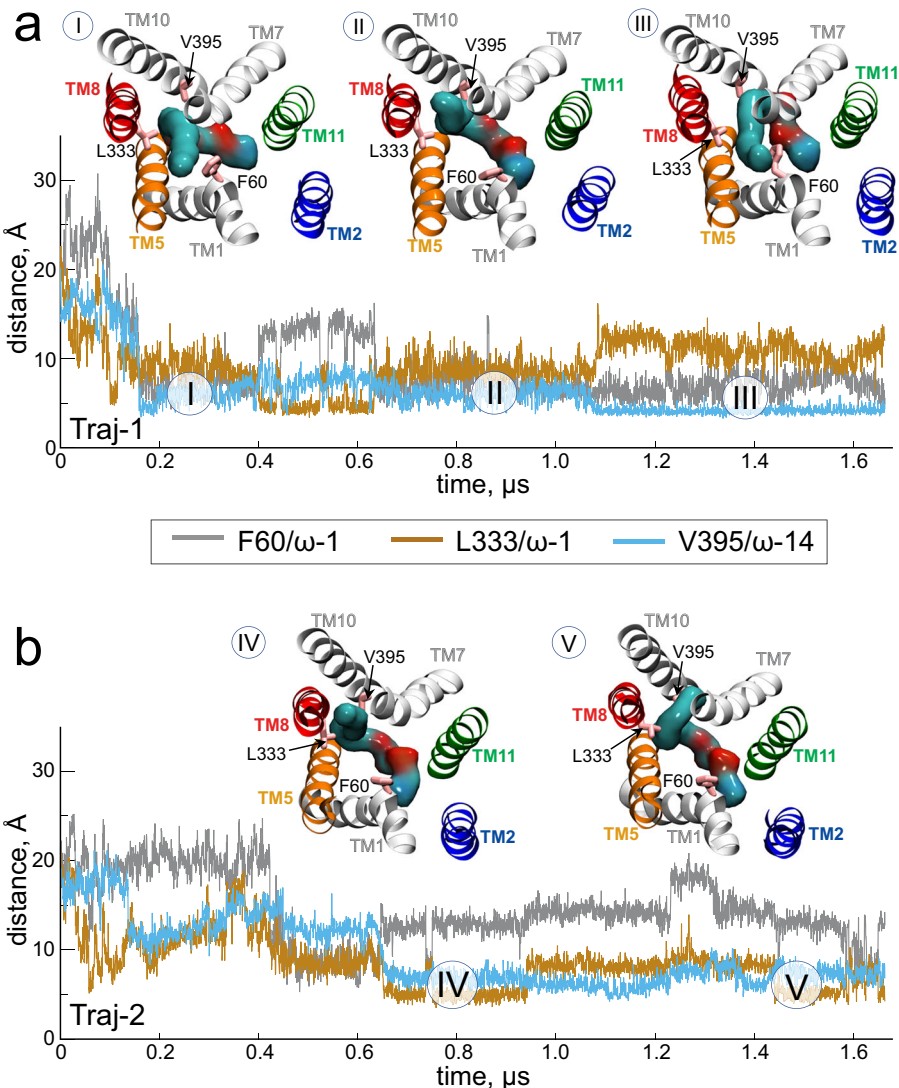

**Fig. 5 | The LPC-18:1 hydrocarbon tail dynamically samples the EC vestibule upon insertion.** Temporal evolution of the distances between the ω−1 carbon of the inserted LPC-18:1 hydrocarbon tail and the $C_\alpha$ of both F60 and L333, and between the ω−14 carbon of the inserted LPC-18:1 hydrocarbon tail and the $C_\alpha$ of V395 in **a** Traj-1 and **b** Traj-2. Structural snapshots from the specified trajectory segments (labelled I-V) are shown in the respective panels. LPC-18:1 is shown in surface representation and the protein is represented as in previous figures. Numbering of the carbon atoms in the hydrocarbon tails is as per IUPAC nomenclature.

with high accuracy and enables identification of molecular features of the protein that are major determinants for distinct conformations[30].

To perform this analysis, we extracted the first and last 4000 frames from Traj-1 (outputted every 160 ps) to represent the ensembles of the OFS and OcS conformations, respectively. Each frame was converted into a 2-dimensional rectangular image in which each pixel represents an atom (in sequential order from top left to bottom right) and coloured using a combination of red, green and blue (RGB) values according to the XYZ coordinates of the atom it represents. These visual representations were then used as input for the DNN algorithm for training, validation and testing. The accuracy of the algorithm on the test set was 99% for the OFS class, and 97.6% for the OcS class, resulting in an overall accuracy of 98.3% (Fig. 8a). Throughout the 100 epochs run, the loss of function for the validation set decreased to 0.057 (Fig. 8b), suggesting that the DNN was highly successful in the classification task.

To identify which molecular determinants were most significant for DNN classification, we carried out visual saliency sensitivity analysis[30]. This involves computing the gradient of the neural network's classification score for all pixels in the visual representation of the trajectory frames, i.e., for all atoms of MFSD2A included in the analysis. The larger the gradient for a given pixel (atom), the more attention the neural network pays in making the classification decision.

This saliency analysis identified residues A293, Y294 and K296 located in the IC end of TM7 as the most significant features for classification (saliency cut-off of ≥ 0.6; Fig. 8c–f). Indeed, when the DNN calculations were repeated with the entire A293-K296 region excluded from the input data, the overall accuracy decreased to ~70%, and the loss of function score for the validation set remained above ~0.5, indicating poor convergence and performance of the algorithm (data not shown). This analysis suggests that classification of the OFS and OcS ensembles was largely influenced by the conformational sampling of this region, and in particular of A293, Y294 and K296 (Fig. 8e, f).

To identify the molecular determinants of this classification in the context of the state-to-state transitions in ggMFSD2A, we compared the conformations of the 293–296 region of TM7 and residues residing in the IC ends of neighbouring TMs in the OFS and OcS. In the OFS, Y294 forms π-π interactions with F427 and Y431 from TM11 (Y294-F427

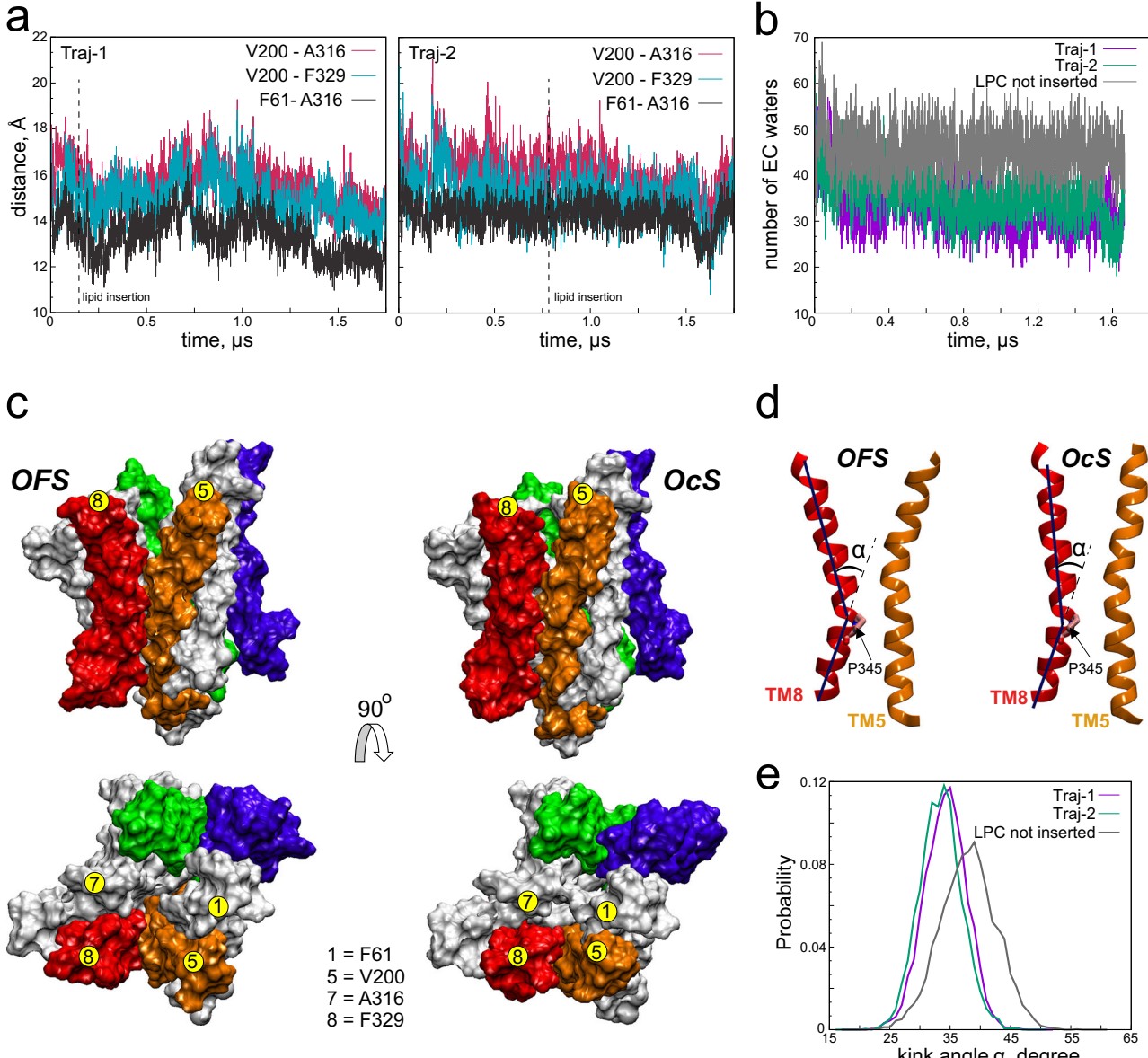

**Fig. 6 | LPC-18:1 insertion leads to an occluded state (OcS) of ggMFSD2A. a** Time-evolution of $C_\alpha$-$C_\alpha$ distances between the EC ends of TM5-TM7 (V200-A316), TM5-TM8 (V200-F329) and TM1-TM7 (F61-A316) in Traj-1 (left) and Traj-2 (right). Dashed vertical lines represent the time-points at which the substrates penetrating via TM5/TM8 in each trajectory engaged in Na⁺-mediated interactions with E312. **b** Number of water molecules in the EC vestibule during Traj-1, Traj-2 and a representative trajectory in which no LPC insertion was observed. Water molecules were classified as belonging to the EC vestibule if any of its atoms were within 3 Å of the following sidechains: Q52, C56, F61, L81, R85, T189, T193, G197, E312, A316, F329, L333, M337, S439, L443 and S446. **c** The OFS (left) and OcS (right) of ggMFSD2A in the plane of the membrane (top) and from the EC milieu (bottom). The protein is represented as in previous figures with yellow circles representing the locations of the four residues described in panel **a**. **d** Ribbon representations of TM5 and TM8 showing the kink angle at P345 (α), in the OFS and OcS in TM8. Residue P345 is highlighted in stick representation. **e** Distribution of the kink angle at P345 (α) in the three trajectories.

and Y294-Y431 distances of 2.6 Å and 2.8 Å, respectively, Fig. 4). In the OcS state, these stacking interactions are broken as Y294 rotates away from F427/Y431 and the IC end of TM7 shifts away from TM11 (Fig. 9b–e). We hypothesised that these conformational rearrangements could mechanistically enable structural transition of the protein from the OcS to the IFS. Indeed, further structural analysis of this IC region revealed that as the transporter transitions from the OcS to the IFS, TM11 moves away from TM2 and towards TM7, so that TM11 and TM7 are again juxtaposed, as in the OFS (Fig. 9c). Intriguing rearrangements of F431 and W403 were also observed upon this transition. While in the OFS and OcS F431 points towards TM7 and W403 points towards the central region (Fig. 9a, b), these residues switch positions in the IFS such that F431 occupies the space previously filled by W403,

and W403, in turn, flips away from the central region, thereby allowing it to associate with other residues that form the functional IC gate (Fig. 9c). These results reveal a mechanistic link between conformational shifts of the IC end of TM7 that occur upon transition between the OcS and IFS, and formation of the IC gate through which substrate is later released into the inner leaflet of the membrane[14].

**Dynamic allosteric coupling between the IC ends of TM7/TM11 and substrate binding site**

Intriguingly, our data demonstrate that rearrangement of the IC ends of TM7/TM11 temporally coincides with insertion of the headgroup-proximal portion of the substrate hydrocarbon tail between TM7 and TM10 at ~1 μs (Figs. 5a and 9e). This observation hints at a potential

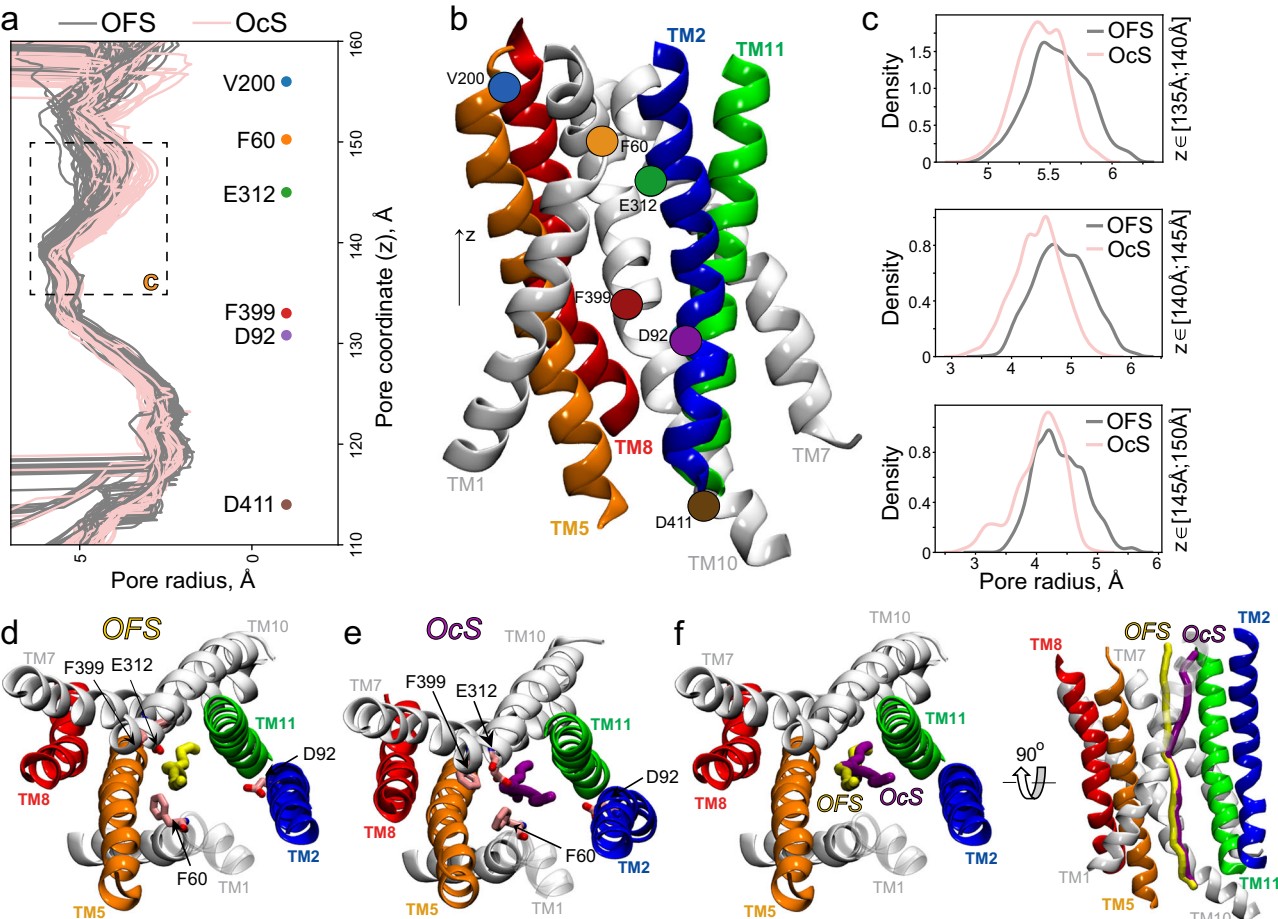

**Fig. 7 | Water distribution in MFSD2A varies between the OFS and OcS. a** Pore radius in angstroms (Å) as a function of pore coordinate (z) calculated using the HOLE software for trajectory frames representing OFS (grey) and OcS (pink). For each ensemble, 100 trajectory frames were considered, and the results for each frame are shown as a separate trace. The locations of several residues along the pore coordinate are shown for reference. The dashed rectangle indicates the region where the pore radius between the OFS and OcS diverges. **b** Representative model of ggMFSD2A in the OFS (only selected TMs are shown, as indicated) highlighting positions of the reference residues from panel **a**. The pore direction was set along

the membrane normal z-axis. (**c**) Distribution of pore radius values for the OFS (grey) and OcS (pink) systems at pore coordinate intervals of z∈[135Å ;140 Å] (top), z∈[140Å ;145 Å] (middle), and z∈[145Å ;150 Å] (bottom). **d, e** EC view of ggMFSD2A in the OFS (**d**) and OcS (**e**) showing the average position of the pore direction in the two ensembles (yellow and purple sticks) respectively. Residues F60, D92, E312 and F399 are highlighted as sticks for reference. **f** EC view and side view of ggMFSD2A in the OFS showing the divergence of pore directions in the OFS and OcS conformations from panels **d** and **e**.

allosteric relationship between the IC ends of TM7/TM11 and residues in the central region/EC vestibule which coordinate the substrate headgroup and tail, respectively (Figs. 3 and 5). To determine if such allostery indeed exists and to quantify its strength, we employed the N-body Information Theory (NbIT) approach to calculate the coordination information (*CI*) between various sites (Supplementary Fig. 1). *CI* quantifies the amount of information that is shared between two sites of the protein called the 'transmitter' and the 'receiver'[31,32]. We defined the transmitter site ($T_S$) as the IC ends of TM7/TM11 (Y294/F427/Y431), while receiver sites R1 and R2 were comprised of residues from the EC vestibule (F61/V200/F329) and central region (E312/F399), respectively (Supplementary Fig. 1a).

The *CI* values between $T_S$ and both R1 and R2 are greater in the OcS (last 4000 frames of Traj-1) than in the OFS (first 4000 frames of Traj-1; Supplementary Fig. 1b). The largest *CI* increase occurred between $T_S$ and R2, which more than doubled in the OcS compared to the OFS. To assess the significance of these observations, we conducted additional NbIT calculations with *receiver* sites that are located on the protein's periphery and not predicted to be involved in substrate binding (R3-R5; Supplementary Fig. 1a). As anticipated, the average *CI* values between $T_S$ and R3-R5 were relatively low (Supplementary Fig. 1b),

demonstrating that the allosteric coupling between these sites is weaker than that between $T_S$ and R1/R2. Notably, while the *CI* between $T_S$ and R3 increased from "low" in the OFS to "average" in OcS, it still remained significantly weaker than the *CI* between $T_S$ and R2 (Supplementary Fig. 1b).

As a negative control, we also performed NbIT analysis on a trajectory where no substrate penetration occurred. Here, the protein remained in the OFS throughout the trajectory and the *CI* values between $T_S$ and R2-R5 in this system were similar to those observed in the OFS ensemble from Traj-1 (i.e., before substrate penetration; Supplementary Fig. 1a). These findings demonstrate a strong allosteric coupling between $T_S$ and R2 in OcS but not in OFS.

To identify allosteric communication channels between the intracellular ends of TM7/TM11 ($T_S$) and the lysolipid headgroup binding site (R2), we then quantified the mutual coordination information (*MCI*) between $T_S$ and R2. *MCI* measures the coordination information shared between the transmitter and receiver sites that is also shared by another site (i.e., a communication channel) in the protein. *MCI* values for the OcS trajectory were high in various structural locations, indicating strongly correlated fluctuations throughout the protein, whereas *MCI* values for the OFS trajectory were

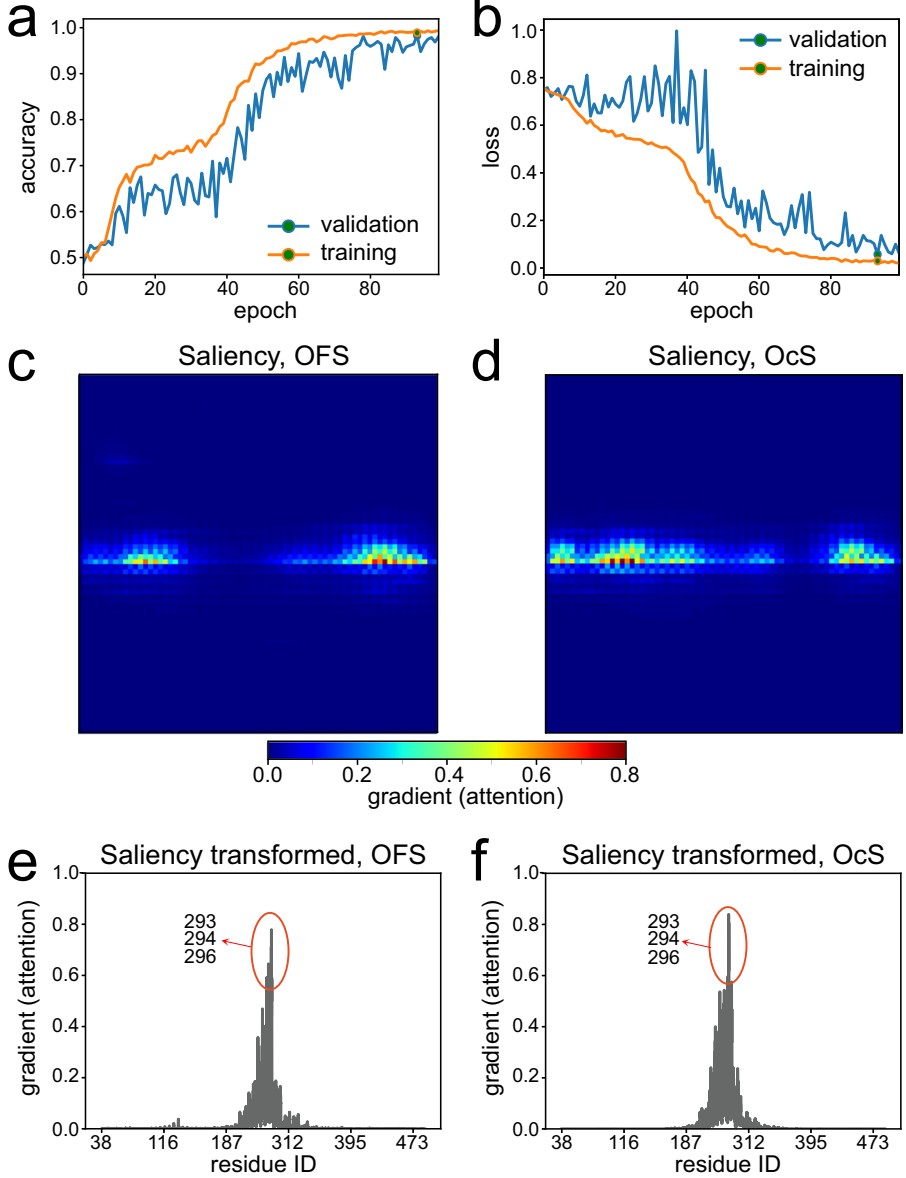

**Fig. 8 | Machine learning analysis identifies the IC end of TM7 as critical for classification of the OFS and OcS. a** Accuracy of the DNN model as a function of epoch and **b** changes in the loss function per epoch for the validation and training sets. DNN-calculated average 2D saliency (attention) maps for the **c** OFS and **d** OcS classes of the 4000 attention maps (for each class). Each pixel represents a TM-region atom within ggMFSD2A (5209 total), organised from the lowest (top left corner) to the highest (bottom right corner) indexed atom. Colour indicates the gradient of each pixel. Pixels with the highest gradient correspond to the most important atoms for classification. Histogram representation of the gradients for all TM-region atoms in the (**e**) OFS and (**f**) OcS classes (**e**, **f**). Residue IDs corresponding to every 1000th indexed atom are indicated on the x-axis. The most significant residues for the classification, A293, Y294 and K296 are highlighted.

substantially lower (Supplementary Fig. 1e). Together, these results demonstrate quantitatively that dynamic changes at the intracellular end of TM7/TM11 ($T_s$) regulate the coordinated motions of F399/E312 (R2), which in turn coordinate the substrate headgroup within the central region of MFSD2A.

## Discussion

Highly enriched within endothelial cells of the blood-brain and blood-retinal barriers, MFSD2A performs $Na^+$-dependent uptake of DHA in the form of a zwitterionic lysolipid into the brain and eyes, respectively[1]. Recent reports have determined high resolution structures of MFSD2A in various states and offer valuable mechanistic insights into how the substrate is released from the protein into the inner leaflet of the membrane to complete the transport cycle[14]. However, the molecular details regarding substrate entry and binding

on the extracellular side of the protein, and the role $Na^+$ plays in this process have not been elucidated. It has also not been demonstrated how the protein transitions throughout the transport cycle, and what molecular determinants facilitate such isomerization.

Taking the results from this study together with recent structure/function studies of MFSD2A[14,16,17], we propose that MFSD2A mediates lysolipid transport using a "trap-and-flip" adaptation of the "rocker-switch" mechanism[13,27]. According to this, the transporter begins in the OFS where the substrate partitions into the MFSD2A central region from the outer leaflet of the membrane via lateral openings between TM5/TM8 and TM2/TM11, while $Na^+$ enters from the extracellular milieu. Once substrate is within the central region, the lateral openings narrow, and the protein contracts – thereby allowing an OcS to be adopted with substrate and $Na^+$ stably bound. Here, the phosphate of the substrate's headgroup is engaged in $Na^+$-bridged interactions with

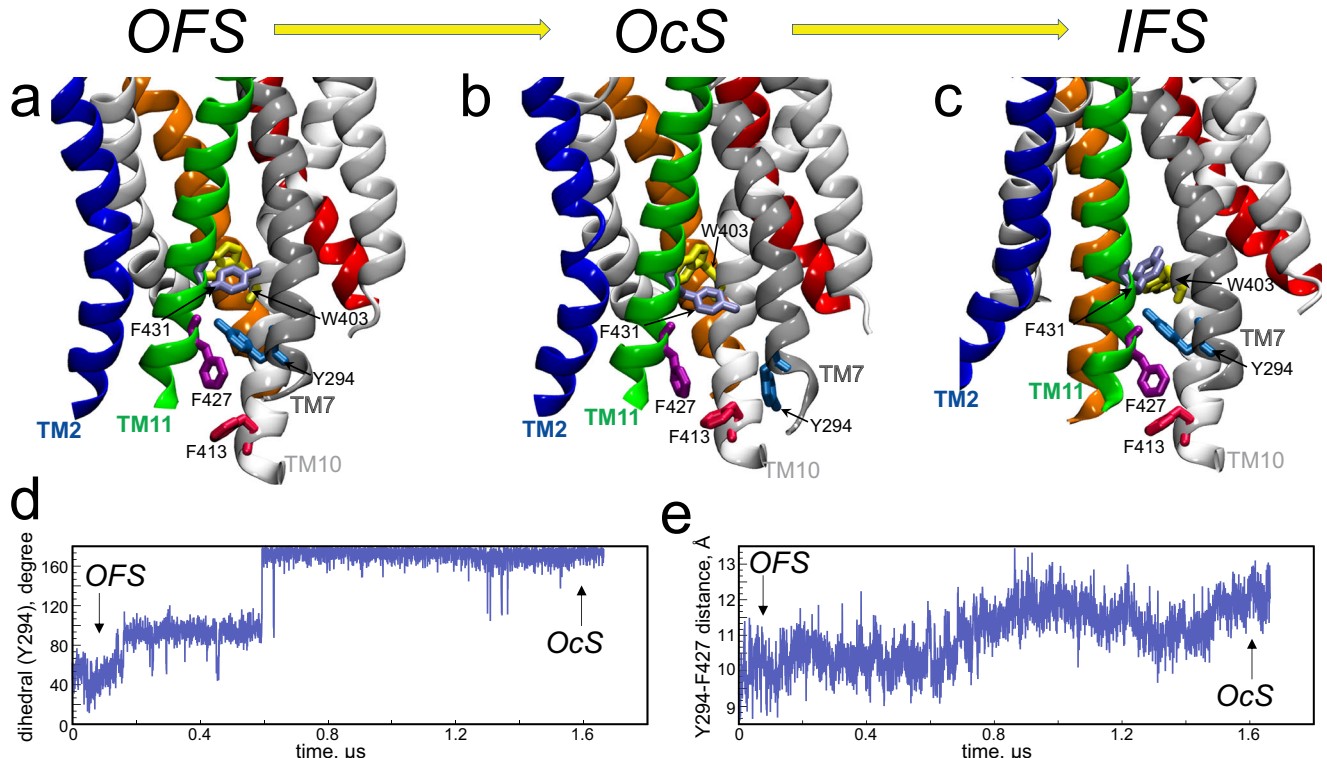

**Fig. 9 | Rearrangement of an aromatic network during state-to-state transitions.** The Y294 neighbourhood of ggMFSD2A in the **a** OFS, **b** OcS and **c** IFS. The IFS is the cryo-EM structure (PDBID: 7MJS)[14], whereas the OFS and OcS structures are the timepoints in Traj-1 simulation shown in panels **d** and **e**. The protein is represented as in previous figures but with TM7 in a darker shade of grey for visual clarity and relevant residues highlighted as coloured sticks. Time evolution of the **d** C-Cα-Cβ-Cγ ($\chi_1$) dihedral angle of Y294 and **e** distance between Y294$_{C\alpha}$ and F427$_{C\alpha}$ in Traj-1. Timepoints defined as the OFS and OcS are indicated.

E312, while the tail is stabilized by a network of aromatic residues including F329, F60 and F61 (Fig. 10a; Substrate Binding Site 1). Notably, only substrates that entered via TM5/TM8 were able to bind in this mode where the substrate is entirely engulfed by the protein and all coordinating residues are both conserved and critical for transport[14]. We hypothesise that movement of Na+ from E312 to D92 then destabilises the bound substrate, thereby causing it to slide down to its second binding site (Fig. 10b; Substrate Binding Site 2) which we previously determined using cryo-EM[14]. These events coincide with the transition of the protein to the IFS, after which the IC gate (F399/W403/M182) opens, and the substrate is released into the inner leaflet of the membrane via the TM5/TM8 lateral opening. These results are consistent with those of our previous study which we demonstrated using a combination of experimental and computational approaches[14].

This mechanism gains further support through various experimental studies. Indeed, our finding that the lateral openings between TM5/TM8 and TM2/TM11 in the OFS provide entry pathways for substrates aligns with two independent experimental investigations[16,33]. Furthermore, the mechanism we propose is in line with that recently suggested for the structurally related MFS glycolipid exporter, LtaA. Much the same as the collective evidence for MFSD2A, LtaA appears to recruit its substrate via the TM5/TM8 lateral opening, binds it in a completely occluded fashion, and uses "rocker-switch" conformational changes to enter the OFS where the substrate is then released via the extracellular TM5/8 and TM2/11 lateral gateways[27]. These similarities suggest that "trap-and-flip" adaptations of the "rocker-switch" mechanism may be a common trait amongst MFS transporters that have adapted to transport lipidic substrates.

Our analyses have revealed that the occlusion of MFSD2A is facilitated by a coordinated movement led by specific transmembrane helices, including TM1, TM2, TM5, TM7, TM8 and TM11. These helices have all have been previously reported as important contributors in the function-related structural transitions in the MFS transporters for small polar molecules, such as the lactose/H+ symporter LacY[15]. Our machine learning analysis has revealed that aromatic residues located within the IC segments of TM7 and TM11 serve as allosteric switches that direct state-to-state transitions in MFSD2A. More specifically, in the OFS and IFS, the IC segments of TM7 and TM11 are engaged with each other via π-stacking interactions, while in the OcS, TM7 distances itself from TM11 (Fig. 9). Notably, in Traj-1 the separation of TM7 from TM11 on the IC side upon transition from the OFS to the OcS temporally coincides with insertion of the headgroup-proximal portion of the substrate hydrocarbon tail between the central segments of TM7 and TM10 (Figs. 9e and 5a respectively; ~1 μs), manifesting in strong allosteric coupling between the IC ends of TM7/TM11 and the lysolipid headgroup binding site (Supplementary Fig. 1). This embedding forces TM7 to tilt such that the EC portion of TM7 moves inward while the IC portion moves outward, away from TM11 (Fig. 6a, c). This suggests that the stabilized insertion of the substrate hydrocarbon tail on the EC side is allosterically connected to the movement of TM7 away from TM11 on the IC side, which may in-turn serve as a preparatory step that allows TM11 the flexibility to move back towards TM7 and away from TM2 as the protein transitions towards the IFS. Interestingly, the re-engagement of TM7 and TM11 upon transition to the IFS directly displaces W403 – a key component of the IC gate – such that it swings away from the central region and towards the area between TM5 and TM8, thereby placing it in the correct position to function as the IC gate.

Taken together, our work provides comprehensive mechanistic insights of MFSD2A-mediated transport and proposes a model for how MFSD2A utilises elements of both "rocker-switch" and "trap-and-flip" mechanisms to mediate lysolipid transport. Still, several questions remain. For example, is either of the two lateral entrances (i.e., between TM5/TM8 or TM2/TM11) preferred under physiological

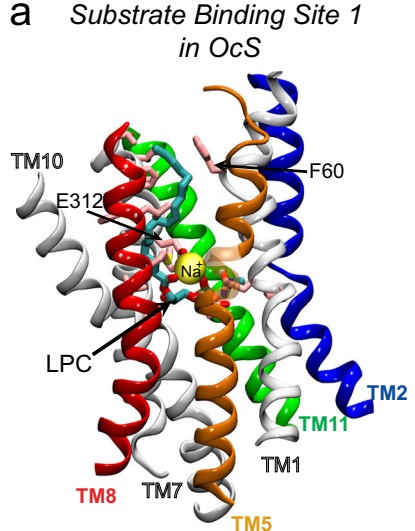

a   *Substrate Binding Site 1 in OcS*

b   *Substrate Binding Site 2 in IFS*

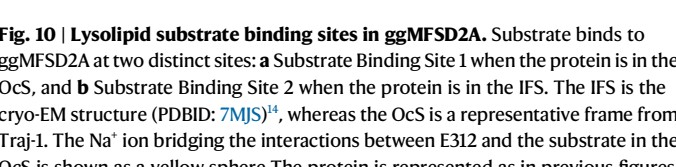

**Fig. 10 | Lysolipid substrate binding sites in ggMFSD2A.** Substrate binds to ggMFSD2A at two distinct sites: **a** Substrate Binding Site 1 when the protein is in the OcS, and **b** Substrate Binding Site 2 when the protein is in the IFS. The IFS is the cryo-EM structure (PDBID: 7MJS)[14], whereas the OcS is a representative frame from Traj-1. The Na$^+$ ion bridging the interactions between E312 and the substrate in the OcS is shown as a yellow sphere The protein is represented as in previous figures

with residues within 3 Å of substrate (LPC-18:1 in the OcS and LPC-18:3 in the IFS, both denoted by "LPC") highlighted. These residues are: Y51, Q52, F60, M309, E312, F315, A316, L317, T320, F329, L333, I336, M337, V395, A398, T435 in the OcS, and M182, V186, M309, L311, E312, I336, M337, I344, Q348, V395, F399, L400 and W403 in the IFS.

conditions? What mechanistic details underpin the precise temporal relationship between Na$^+$ and substrate headgroup binding at E312 and between Na$^+$ binding at E312 and D92? What are the molecular mechanisms underlying the OcS to IFS transition and movement of the substrate from Site 1 to Site 2 (Fig. 10)? Furthermore, what role does a conserved disulphide crosslink – which bridges an EC loop of the N-domain to one of the C-domain – play in substrate selectivity and/or transporter dynamics? Lastly, our research has not addressed the potential impact of membrane lipid composition, including cholesterol, sphingomyelin, or highly charged inositol lipids, on the function of MFSD2A, despite their known ability to modulate the conformational dynamics of membrane proteins. While further studies will be needed to address these remaining questions, our findings here have provided substantial steps towards understanding the mechanistic basis underpinning MFSD2A-mediated lysolipid transport.

## Methods

### Molecular constructs for molecular dynamics simulations

All the computations described here were based on the outward-facing cryo-EM structure of mmMFSD2A (PDBID: 7N98)[16]. This structure was used to construct a homology model of an outward-facing full-length ggMFSD2A. We chose to study ggMFSD2A since our previously published computational work on the inward facing MFSD2A protein structure was based on it[14]. Thus, consideration of the same protein in this work allowed us to compare more conveniently the current results to those published in ref. 14. We note that mmMFSD2A and ggMFSD2A structures are expected to be similar as their sequences are highly homologous (74% identity)[14].

To build an outward-facing full-length model of ggMFSD2A, the OFS mmMFSD2A structure (PDBID: 7N98)[16] and the predicted structures of the N- and C-domains from the IFS (PDBID: 7MJS)[14] were used as structural templates. Using Modeller v9[34], 100 models of the full-length ggMFSD2A were generated and ranked according to the molpdf energy score. The model with the lowest molpdf, which can be found at https://zenodo.org/record/7933371#.ZF_jKOzMLQ0 repository, was chosen for all the subsequent computational experiments. To create E312D mutant ggMFSD2A construct, the mutation was introduced to

the models of the WT protein using CHARMM-GUI web server[35] (see also "Results" section). Protonation states of the titratable residues were predicted at pH 7 with Propka 3.1[36], resulting in default protonation configurations.

### Atomistic MD simulations of the wild type ggMFSD2A in a pure phosphatidylcholine membrane

The OFS ggMFSD2A model was embedded into a membrane containing 484 POPC lipids. The protein-membrane complex was then immersed into a solution box containing 52,500 water molecules, 144 K$^+$ ions, and 145 Cl$^-$ ions (0.15 M ionic concentration). The final simulation box had a size of ~135Åx135Åx135Å and contained ~231,000 atoms, including explicit hydrogens.

The assembled system was subjected to a short equilibration run with NAMD 2.13[37] using a standard set of equilibration scripts provided by CHARMM-GUI. After this initial equilibration, the velocities of all the atoms were randomly regenerated and the system was subjected to extensive multi-replicate MD simulations whereby the system was simulated in 48 independent replicates, each replicate 1.4-1.7 μs long (~74 μs cumulative sampling). These MD simulations were carried out with OpenMM 7.4[38] and implemented Particle Mesh Ewald (PME) for electrostatic interactions. The runs were performed at 310 K, under isothermal-isobaric (NPT) ensemble conditions using semi-isotropic pressure coupling, and with 4 fs integration time-step (with mass repartitioning). MonteCarloMembraneBarostat and Langevin thermostats were used to maintain constant pressure and temperature, respectively. Additional parameters for these runs included: "friction" set to 1.0/ps, "EwaldErrorTolerance" 0.0005, "rigidwater" True, and "ConstraintTolerance" 0.000001. The van der Waals interactions were calculated applying a cut-off distance of 12 Å and switching the potential from 10 Å.

### Atomistic MD simulations of WT ggMFSD2A in a POPC/LPC-18:1 membrane

Since MFSD2A transports lysolipids, such as LPC-18:1[2], we sought to investigate how these substrates spontaneously penetrate and bind MFSD2A using MD simulations. To this end, we constructed a system

with externally placed LPC-18:1 by replacing the 15 POPC lipids on the extracellular leaflet that were closest to the protein in the trajectory frame after the initial CHARMM-GUI equilibration phase (described above) with 15 LPC-18:1 molecules. This was achieved by aligning the lysolipid model with the selected POPC lipids (using the phosphorus atoms of the two lipids as the reference group) and then removing the POPC lipids from the system. The LPC-18:1-containing system was minimised (for 3000 steps) and subjected to a short equilibration (120 ps long). After this, the velocities of all the atoms were randomly regenerated and the system was simulated in 48 independent replicates, each replicate 1.6–1.7 μs long, resulting in ~79 μs cumulative sampling (half of the replicates were run for 1.6 μs and the other half of the replicates – for 1.7 μs). These multi-replicate MD simulations were performed using OpenMM 7.4 with the same run parameters described above implemented.

## Sampling of an OcS of WT ggMFSD2A in a POPC/LPC-18:1 membrane

To sample the dynamics of WT ggMFSD2A OcS in a POPC/LPC-18:1 membrane, we identified from the above-described MD simulations a trajectory frame in which the transporter was occluded around one LPC-18:1 molecule inserted deep into the protein (see "Results" section). This conformation was used as the starting point for additional sets of multi-replicate MD simulations in which the system was run in 24 replicates, each replicate ~2.1 μs long (~50 μs cumulative sampling). These MD simulations were performed using OpenMM 7.4 with the same run parameters described above implemented.

## Atomistic MD simulations of E312D ggMFSD2A in a POPC/LPC-18:1 membrane

E312D ggMFSD2A was embedded into the POPC/LPC-18:1 bilayer described above. The system was minimised (for 3000 steps) and subjected to a short equilibration (120 ps long). After this, the velocities of all the atoms were randomly regenerated and the mutant system was simulated in 24 independent replicates, each 1.5 μs long (~36 μs cumulative sampling). These multi-replicate MD simulations were also performed using OpenMM 7.4 with the same run parameters described above implemented.

For all simulations we used the latest CHARMM36m force-field for proteins and lipids[39], as well as recently revised CHARMM36 force-field for ions which includes non-bonded fix (NBFIX) parameters[40]. For all the standard MD data analyses and visualisation VMD software version 1.9.3 was used[41].

## Machine learning analysis of the MD data

To identify structural elements in ggMFSD2A that sample the most divergent conformations between the OFS and OcS ensembles, we used the Deep Neural Network (DNN) classification algorithm described previously (https://github.com/weinsteinlab/DNN)[30]. Briefly, 4000 trajectory frames (with the stride of 160 ps) representing the OFS ensemble and 4000 frames (with the stride of 160 ps) representing the OcS ensemble of states were selected (see Results). For both sets of data, the trajectory frames were positionally and orientationally scrambled[30] and the coordinates of the TM segments (residue 39–68, 75–101, 110–129, 137–167, 171–202, 233–263, 292–318, 328–353, 358–374, 382–416, 424–450 and 465–491) were extracted and saved into separate trajectory files. Each trajectory frame was then converted into a visual representation suitable for input to the DNN[30] (see "Results" section).

The DNN was constructed as a Densely Connected Neural Network with 4 dense blocks containing 6, 12, 36 and 24 layers respectively, 96 initial filters, a growth rate of 48 filters per layer, and a reduction ratio of 0.5, in Keras[42] with a Tensorflow[43] backend based on an established implementation[44]. The 8000 trajectory frames were randomly split into a training, validation, and test sets using the ratio of

56:24:20, respectively. The neural network was trained on the training and validation sets and tested on the test set. The sensitivity analysis was performed by computing the gradient using the visual saliency package provided by keras-vis[45] with guided backpropagation. For more details of the algorithm, see ref. 30.

## Helical kink analysis

To analyse proline kink distortions, we used ProKink[46] tool in the publicly available software Simulaid[47].

## Water pore analysis

To compare water distribution within the protein in the OFS and OcS, we carried out pore analysis on the 4000-frame OFS and OcS trajectories described above using the programme HOLE (http://www.holeprogram.org/)[48]. For each ensemble, HOLE was run on 100 regularly spaced frames (i.e., on every 40th frame) which were aligned on the TM segments to remove positional and translational degrees of freedom. Due to their high flexibility, the N- and C-terminal segments (residues 1–38 and 492–528, respectively) were not considered for this analysis. The pore was specified to lie along the membrane normal z axis, and the initial point for building the pore represented the (x, y, z) coordinates of the F399 $C_\alpha$ atom.

## N-body Information Theory analysis

To quantify the allosteric coupling between various sites on ggMFSD2A, we applied N-body Information Theory (NbIT) analysis (https://github.com/weinsteinlab/NbIT)[31] to the trajectories of the OFS and OcS ensembles described above, as well as to the representative trajectory in which no substrate penetration into the protein core was observed (i.e., the system was in the OFS state throughout the trajectory, see Results). Briefly, configurational entropy $H$ was calculated from the covariance matrix ($\mathbf{C}$) of all atomic positions ($\mathbf{X}$) in the protein:

$$H(X) = \frac{1}{2}\ln|2\pi e \mathbf{C}(\mathbf{X})| \tag{1}$$

From the above, total correlation ($TC$) was then quantified as:

$$TC(X_1, \ldots, X_N) = \sum_{i=1}^{N} H(X_i) - H(X_1, \ldots, X_N) \tag{2}$$

where $X_i$-s represent components of 3N-dimensional vector corresponding to x, y, z atomistic coordinates of the atoms in the set, and $H(X_1, \ldots, X_N)$ is the joint entropy of the set. Using $TC$, coordination information ($CI$) was obtained as:

$$CI(\{X_1, \ldots, X_N\}|X_m) = TC(X_1, \ldots, X_N) - TC(X_1, \ldots, X_N|X_m) \tag{3}$$

In the above, $TC(X_1, \ldots, X_N|X_m)$ represents conditional total correlation between $\{X_1, \ldots, X_N\}$, conditioning on $X_m$. From $CI$, the mutual coordination information (MCI) was calculated as:

$$MCI(\{X_1, \ldots, X_N\}|X_m, X_n) = CI(\{X_1, \ldots, X_N\}|X_m) + CI(\{X_1, \ldots, X_N\}|X_n) - CI(\{X_1, \ldots, X_N\}|X_m, X_n) \tag{4}$$

NbIT was applied to all the non-hydrogen atoms of ggMFSD2A. The IC ends of TM7/11, (residues Y294, F427 and Y431) were defined as the *transmitter site* 'T$_s$'[31]. *Receiver sites* 'R1' (F61/V200/F329), 'R2' (E312/F399), 'R3' (P119/V122/I254), 'R4' (S366/A367/A390) and 'R5' (C39/C43/C256) were tested for allosteric coupling with T$_s$. R1 and R2 are comprised of residues that coordinate the substrate's hydrocarbon tail and headgroup respectively, while R3-R5 are located on the periphery of protein and distal from T$_s$, R1 and R2. Residues within R3, R4 and R5

maintained stable interactions with other residues in the same receiver sites in both the OFS and OcS, and thus these sites are not expected to be strongly allosterically coupled to the $T_s$, R1 or R2.

*CI* data was clustered using a Fisher-Jenks algorithm into three groups of sites with various levels of allosteric coordination: low ($CI < 15\%$), average ($15\% \leq CI \leq 28\%$) and high ($CI > 28\%$) as previously described (https://github.com/mthh/jenkspy)[32].

## Reporting summary

Further information on research design is available in the Nature Portfolio Reporting Summary linked to this article.

## Data availability

The homology model of chicken MFSD2A in the outward facing state, as well as the trajectory files from Supplementary Movie 1 and Movie 2 are freely available via Zenodo public repository: https://zenodo.org/record/7933371#.ZF_jKOzMLQ0. Requests for all the additional structural models and MD simulation trajectories should be directed to and will be fulfilled by the corresponding author. Any additional information required to re-analyse the data reported in this paper is available from the corresponding author upon request.

## Code availability

The NbIT and the DNN codes used to perform allostery and machine learning analyses described above are publicly available at the following github repositories: https://github.com/weinsteinlab/NbIT and https://github.com/weinsteinlab/DNN, respectively.

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

## Acknowledgements

We are grateful to Dr. Derek Shore for his help to establish a public repository for the inhouse DNN analysis tool described in this work. G.K. gratefully acknowledges support from the 1923 Fund. F.M. was funded by NIH grant R21MH125649. R.J.C. was supported by the Simons Society of Fellows (award number 578646). The authors acknowledge resources of the Oak Ridge Leadership Computing Facility (INCITE allocation BIP109) at the Oak Ridge National Laboratory, which is supported by the Office of Science of the U.S. Department of Energy under contract no. DE-AC05-00OR22725. The authors gratefully acknowledge the enabling role of Chris Carothers, director of the Center for Computational Innovations (CCI) at the Rensselaer Polytechnic Institute (RPI), for the kind help and support of efficient and sustained access to the AiMOS supercomputer at CCI. These resources were generously awarded through the COVID-19 High Performance Computing Consortium. The authors gratefully acknowledge the use of resources of the Oak Ridge Leadership Computing Facility, which is a DOE Office of Science User Facility supported under Contract DE-AC05-00OR22725, and of in-house computational resources of the David A. Cofrin Center for Biomedical Information in the Institute for Computational Biomedicine at Weill Cornell Medical College.

## Author contributions

G.K., R.J.C. and F.M. conceived the project. G.K. designed the study, carried out MD simulations, and, together with S.B. performed majority of the analyses. A.P. and S.B. performed the machine learning analysis. G.K., R.J.C. and F.M. wrote the manuscript. All authors contributed to the interpretation of the data and editing the manuscript.

## Competing interests

The authors declare no competing interests.
