## [Peer Review File · Nature Communications]

Substrate binding-induced conformational transitions in the omega-3 fatty acid transporter MFSD2AREVIEWER COMMENTS

Reviewer #1 (Remarks to the Author):

Khelashvili et al. propose a study where they wish to address three mechanistic questions concerning transport by the MFSD2A transporter. The study addresses the questions of how substrates enter and bind to MFSD2A on the extracellular side, as well as the role of Na⁺ in substrate binding, and how substrate binding, at large, facilitates isomerization. The major contribution of this work involves the generation of a large dataset from classical MD simulations on a very large system (231k atoms, and ~240 μs of simulation across a variety of different system constructs). The dataset was generated by means of several independent trajectories being run from the same starting point but with different velocities, what the authors refer to as “cumulative sampling”. From this dataset, a deep learning-based software was applied to generate a list of amino acid sites believed to be critical for promoting transition from outward-facing (OF) to occluded (OC) states of MFSD2A. Furthermore, the authors also emphasize the roles of Na⁺ ion and the E312 site in facilitating lipid recognition along the extracellular vestibule. MFSD2A is a transporter which has appeared in a few high-profile papers, making the transporter worthwhile for modeling and simulation study.

Concerning methodology, the authors promote the use of machine learning analysis to suggest what residues are important for substrate molecular recognition and transporter conformational change. However, the execution of such a methodology is not novel. In the MD simulation community, clustering analysis (which are forms of unsupervised machine learning) and Data-driven modeling has also been used for similar purposes (<https://elifesciences.org/articles/60715#fig4>). The authors are using a methodology which they had published in their 2019 Molecules paper. The application of this method in the paper does not provide any methodological novelty. This method is a fine approach for discerning important residues with minimal simulation input, but it is not the only way by which the authors could better understand the mechanism of this transporter.

The other part of the equation is adequately describing a novel mechanism. While Kelashvili et al. portray that the “trap-and-flip” mechanism may be perceived as the dominant mode of substrate transport, their claims are not entirely irrefutable. The authors do not strongly advocate for the trap-and-flip mechanism in their results/discussion. This is because, by definition, either the trap-and-flip or the credit-card mechanism describe substrate transport, not just substrate binding. That is, for a definitive conclusion on what mechanism should be seen, this study requires alternate access (transition from OF -> OC -> inward-facing, IF) to be observed through simulation or approximated based on clever system setup. From their movies of Traj-1 and Traj-2, the authors show that characteristics from both types of mechanisms can be seen as possibly sampled rare events. However, in neither case do the authors observe complete alternate access. One can consider that if Supplementary Video 2 involved just the lipid entering between TM2/TM11, that this lipid could have proceeded to the occluded state through what looks like a credit-card mechanism. There is not enough definitive evidence to suggest

exactly that one mechanism occurs. Thus, the authors do satisfy their research question of describing substrate binding, but characterizing substrate binding alone without alternate access is insufficient for characterizing the mechanism of transport.

There are a few strategies the authors can perform to validate one mechanism over the other.

Strategy 1. Assuming the authors continue from their OF-bound structures, the authors can:

- Use the Traj-1 data as a starting point for a biased “trap-and-flip” simulation. Make sure that the rest of the lipid bilayer is POPC; that no other lipids will enter the protein.
- Use the Traj-2 data where only the TM2/TM11 lipid entering is retained and the other lipid is removed. Use this starting conformation to build a new system for evaluating the “credit card” mechanism with a biased simulation. Make sure that the rest of the lipid bilayer is POPC; that no other lipids will enter the protein.

The authors could then promote access to the IF state using a biasing method and see which mechanism offers a lower energy.

Strategy 2. It is a little odd that the authors did not run any simulations from the inward-facing (IF) state considering that a structure is available. The authors can/should perform a similar simulation regime, except this time starting from the IF state. I am not sure whether MFSD2A is a reversible transporter, but I imagine that ion and lipid binding from the intracellular vestibule should result in something like a stabilized occluded state. After all, it is known that MFSD2A uses ion-coupling to stabilize conformational states. Because the authors analyze their data from the perspective of independent trajectories, it makes the sampling of rare events more difficult; however, the classical nature of their simulations suggests that the rare events being observed are likely to have occurred given lowest probability. Thus, the authors should be able to see whether a “trap-and-flip” or “credit card” mechanism is seen for IF binding. In this way, if the authors are able to approximate an occluded state from the IF side, then the overall data contained in the paper could emulate alternate access, thereby enabling the authors to promote one mechanism over the other.

Overall, this study incompletely describes a transport mechanism. The mechanism is important for a system which has gathered a lot of interest from the structural biology field. However, even with a resolved mechanism, the choice of methodology is not rigorous enough (for example there is no reporting of kinetic or thermodynamic information, etc.). With a completely described transport mechanism, I could advocate for publication.

MINOR COMMENTS

-It is quite fascinating how the Deep Neural Network (DNN) classification method can succeed with such little data, as the authors used this method on only 4000 frames per ensemble. It would be interesting to see the authors compare the features identified from the DNN method to other feature classification techniques, such as residue-residue contact score (RRCS; <https://elifesciences.org/articles/50279>). The authors deserve to emphasize the success of their DNN technique in classifying critical residues with minimal need for input data.

-For each of the different ensemble-based results, the independent replicates are of different lengths. Can the authors comment on why the lengths of individual trajectories pooled for cumulative sampling vary? While it is understandable that the lengths of different trajectories can be different for some computing-dependent reasons, but I noticed that different system setups have differing simulation run lengths, which appeared odd. Justifying the reason for why these sets of trajectories are of different length is great for transparency's sake.

-I was not able to understand the rationale behind the LPC/POPC membrane setup until I read further into the paper. The authors should include their reasoning behind the dual component bilayer setup in the Methods section as well.

-Can the authors confirm that they used the MonteCarloMembraneBarostat in their OpenMM simulations?

-The language used in the Discussion is very passive, or rather, not deliberate enough. Specifically, I felt this when it comes to describing what mechanism what exact mechanism – “trap-and-flip” or “credit-card” mechanism. The authors state that “they propose a refined model of MFSD2A-mediated transport that combines elements of both the ‘trap-and-flip’ and ‘rocker-switch’ models.” The trap-and-flip model of transport is not at the same level of mechanistic description as rocker-switch; that is, rocker-switch is a fundamental mechanistic description for alternate access, while trap-and-flip is a description of substrate translocation and not conformational change associated with alternate access. Assuming that a reader reads through this paper quickly or is not as familiar with the transporter field terminology, the writing of these mechanistic terms as is misleading because of the different levels of generality (i.e., suggests transport can be either/or). However, there is some nuance here in that perhaps it is possible that the credit-card mechanism does not strictly adhere to the tenets of alternate access? Again, the authors need to better layout their discussion to contextualize their results in light of transporter alternate access schemes. Rather than just restate the results, the authors should firmly state what the translocation strategy is in clear terms of their results once the major comments have been addressed.

Reviewer #2 (Remarks to the Author):

Khelashvili et al has studied the MFSD2A protein and its lipid penetration/binding process using an extensive set of molecular dynamics simulations. While the volume of the data generated is impressive and the methodology is generally sound, I am a bit hesitant to agree with the authors on the significance of their findings.

1. A number of standard analyses are done. Many other types of analyses could be also helpful and may capture important information (rotational motion of helices, pore size dynamics, water profiles, PCA, dynamic network analysis, etc). The MFS family members have been studied extensively and the current manuscript fails to look at many interesting aspects of the structural dynamics of MFSD2A when compared to some other publications. Although there are certainly a number of interesting observations reported in the current manuscript, I think there is more to do with these trajectories.

2. Although there is no doubt that an extensive set of simulations is performed here, the simulations are still too short and too few to capture longer timescales that may be associated with the entrance and binding of lipids here. For instance, at longer timescales a different pathway and a different binding site may become relevant with a slower kinetics but a favored thermodynamics.

3. An analysis of the pore size and/or water profile along the pore is missing in this study. I think such analyses would be of interest to the readers. Fig. 6 tries to address these indirectly but much better analysis techniques are already established in the literature that can be used.

4. The machine learning technique used in this work reveals the importance of TM7 in conformational changes. Some of the other helices are also mentioned in the study. It is useful to compare these findings to some similar findings in the literature with respect to role of TM helices like 1 and 7 or 5 and 11.

5. The authors use ensemble MD to describe a set of simulations consisting of 24-48 replicates. I do not believe this is an appropriate use of the term ensemble. A statistical ensemble consists of a large number of microstates that form one or more macrostates. An ensemble MD would ideally use millions of replicates or perhaps thousands with the current computation standards. In this case, a few dozens of simulations are performed each for ~ 1-2 us. I think what is more appropriate is multi-replicate MD or something of that sort rather than ensemble MD. Now the microstates that are generated may be called an ensemble or a representative ensemble with some abuse of terminology, which is fine. So I am more concerned with the "ensemble MD" terminology that may imply a much larger number of trajectories.

Reviewer #3 (Remarks to the Author):

In this study, Khelashvili et al use molecular dynamics (MD) to address the mechanism of lysophosphatidylcholine (LPC) transport by MFSD2A at the blood brain barrier. They run MD simulations on a chicken homology model of the outward facing structure to show how the LPC enters the transporter from the outer leaflet. This complements their previous work on the inward facing structure of the chicken protein. They then apply machine-learning based analysis to uncover the allostery between LPC binding and transporter conformational changes. In their analysis, they also highlight interesting effects of the Na⁺ ion on the transport mechanism.

The methodology is sound, the results are convincing, and the manuscript is well written. I found it an interesting read, and think their conclusions are well justified.

I suggest some minor changes which I feel would bolster the study:

- The simulations use a simple membrane model. This is understandable, but it would be worth mentioning in the discussion that a more complex membrane could potentially impact the observations, especially the presence of lipids such as cholesterol or sphingomyelin
- I think an image of LPC vs POPC would be useful to guide the readers
- In Fig 2d, there is a striking difference between LPC and POPC entry into the transporter, especially via the TM5/TM8 route. Is there a physical explanation for this selectivity?
- Related to this, does the additional tail or free OH of LPC make any specific interactions with MFSD2A which could explain the selectivity?
- It would sometimes be useful to see the LPC substrate in more detail (i.e. as sticks). This is especially the case in Fig. 3d, to visualise LPC-LPC contacts and the LPC-Na-E312 bridge in detail. This would make a good additional panel, either to the main figures or supplementary data.
- Were default protonation states used for all side chains? Were any pKa calculations run to confirm this is the case?
- The DNN used here, and previously described by the authors, seems like a very interesting tool. I would encourage them to make the code available if it is not already (I couldn't see it).
- I also request the authors make their chicken homology model available via an open access online repository.

Signed,

Robin Corey

REVIEWER #1:

Khelashvili et al. propose a study where they wish to address three mechanistic questions concerning transport by the MFSD2A transporter. The study addresses the questions of how substrates enter and bind to MFSD2A on the extracellular side, as well as the role of Na⁺ in substrate binding, and how substrate binding, at large, facilitates isomerization. The major contribution of this work involves the generation of a large dataset from classical MD simulations on a very large system (231k atoms, and ~240 μs of simulation across a variety of different system constructs). The dataset was generated by means of several independent trajectories being run from the same starting point but with different velocities, what the authors refer to as “cumulative sampling”. From this dataset, a deep learning-based software was applied to generate a list of amino acid sites believed to be critical for promoting transition from outward-facing (OF) to occluded (OC) states of MFSD2A. Furthermore, the authors also emphasize the roles of Na⁺ ion and the E312 site in facilitating lipid recognition along the extracellular vestibule. MFSD2A is a transporter which has appeared in a few high-profile papers, making the transporter worthwhile for modeling and simulation study.

We thank the Reviewer for their appreciation of our work.

1) Concerning methodology, the authors promote the use of machine learning analysis to suggest what residues are important for substrate molecular recognition and transporter conformational change. However, the execution of such a methodology is not novel. In the MD simulation community, clustering analysis (which are forms of unsupervised machine learning) and data-driven modeling has also been used for similar purposes (<https://elifesciences.org/articles/60715#fig4>). The authors are using a methodology which they had published in their 2019 Molecules paper. The application of this method in the paper does not provide any methodological novelty. This method is a fine approach for discerning important residues with minimal simulation input, but it is not the only way by which the authors could better understand the mechanism of this transporter.

We agree with the Reviewer that there are many alternative approaches to analyze the MD trajectories. Here we chose to use the machine learning-based analysis method that we previously developed as we have found that it performs classification tasks from MD simulation data exceptionally well (Ref 36, Plante *et al.*, *Molecules* 2019). Both this Reviewer and others have acknowledged the method's efficiency and effectiveness in the current study, which enabled us to derive important molecular insights into the transport mechanism. In the interest of transparency, and as per Reviewer three's suggestion, we have made the machine learning code publicly available at the following GitHub repository: <https://github.com/weinsteinlab/DNN>.

We apologize if our language inadvertently suggested that this methodology was newly developed for this study. This was by no means our intention. In fact, contrary to this, we thought we referred to this method throughout the manuscript as a methodology as previously developed/described (lines 196, 407). We hope this clarification resolves any confusion.

2) The other part of the equation is adequately describing a novel mechanism. While Kelashvili et al. portray that the “trap-and-flip” mechanism may be perceived as the dominant mode of substrate transport, their claims are not entirely irrefutable. The authors do not strongly advocate for the trap-and-flip mechanism in their results/discussion. This is because, by definition, either the trap-and-flip or the credit-card mechanism describe substrate transport, not just substrate binding. That is, for a definitive conclusion on what mechanism should be seen, this study requires alternate access (transition from OF -> OC -> inward-facing, IF) to be observed through simulation or approximated based on clever system setup. From their movies of Traj-1 and Traj-2, the authors show that characteristics from both types of mechanisms can be seen as possibly sampled rare events. However, in neither case do the authors observe complete alternate access. One can consider that if Supplementary Video 2 involved just the lipid entering between TM2/TM11, that this lipid could have proceeded to the occluded state through what looks like a credit-card mechanism. There is not enough definitive evidence to suggest exactly that one mechanism occurs. Thus, the authors do satisfy their research question of describing substrate binding, but

characterizing substrate binding alone without alternate access is insufficient for characterizing the mechanism of transport.

There are a few strategies the authors can perform to validate one mechanism over the other.

Strategy 1. Assuming the authors continue from their OF-bound structures, the authors can:
- *Use the Traj-1 data as a starting point for a biased “trap-and-flip” simulation. Make sure that the rest of the lipid bilayer is POPC; that no other lipids will enter the protein.*
- *Use the Traj-2 data where only the TM2/TM11 lipid entering is retained and the other lipid is removed. Use this starting conformation to build a new system for evaluating the “credit card” mechanism with a biased simulation. Make sure that the rest of the lipid bilayer is POPC; that no other lipids will enter the protein.*
The authors could then promote access to the IF state using a biasing method and see which mechanism offers a lower energy.

Strategy 2. It is a little odd that the authors did not run any simulations from the inward-facing (IF) state considering that a structure is available. The authors can/should perform a similar simulation regime, except this time starting from the IF state. I am not sure whether MFSD2A is a reversible transporter, but I imagine that ion and lipid binding from the intracellular vestibule should result in something like a stabilized occluded state. After all, it is known that MFSD2A uses ion-coupling to stabilize conformational states. Because the authors analyze their data from the perspective of independent trajectories, it makes the sampling of rare events more difficult; however, the classical nature of their simulations suggests that the rare events being observed are likely to have occurred given lowest probability. Thus, the authors should be able to see whether a “trap-and-flip” or “credit card” mechanism is seen for IF binding. In this way, if the authors are able to approximate an occluded state from the IF side, then the overall data contained in the paper could emulate alternate access, thereby enabling the authors to promote one mechanism over the other.

Overall, this study incompletely describes a transport mechanism. The mechanism is important for a system which has gathered a lot of interest from the structural biology field. However, even with a resolved mechanism, the choice of methodology is not rigorous enough (for example there is no reporting of kinetic or thermodynamic information, etc.). With a completely described transport mechanism, I could advocate for publication.

We thank the Reviewer for these comments as they highlight some gaps in our explanation of our results, the proposed mechanism, and references to results already published in support of the mechanism we propose.

Firstly, we recognize that we did not clearly delineate the "trap and flip" and "credit card" mechanisms, nor how both mechanisms may necessitate "rocker-switch"-type conformational changes in the protein for transport to occur. We now address these points in the revised manuscript (pages 4-5, 18-20). To reiterate here, the "trap and flip" and "credit card" mechanisms describe two distinct modes of lipidic substrate engagement with the protein during the translocation process. In the "trap and flip" mechanism, the entire substrate (both charged headgroup and hydrophobic tail) enters the protein via the membrane and flips upon entry, becoming inverted and fully enclosed ("trapped") by the protein. In contrast, the "credit card" mechanism involves the charged headgroup of the lipidic substrate entering the protein from the membrane, while the hydrophobic tail(s) remain in the lipid bilayer. Notably, the "trap and flip" and "credit card" aspects describe how the lysolipid substrate enters and binds to the protein, while the "rocker-switch" aspect describes the conformational changes the protein undergoes to accommodate this binding and facilitate transport. The “rocker-switch” aspect of the mechanism is well defined for MFS transporters, and indeed has been confirmed as the mechanism of alternating access for MFSD2A by the independent determination of the protein in both an OFS (Wood *et al.*, *Nature* 2021) and an IFS (Cater *et al.*, *Nature* 2021). The unknown aspect of the mechanism at the onset of our study surrounded how MFSD2A has adapted its substrate engagement mechanism (i.e., via either the “trap and flip” or “credit card” mechanism) to allow for the transport of lysolipid substrates – which is atypical for MFS transporters. We have expanded our explanation of this in the introduction to clarify these distinctions (lines 94-97 and 105-112) and, again, thank the Reviewer for bringing this to our attention.

In the current study we have observed multiple events of substrate penetration from the extracellular leaflet of the membrane into the protein central region via the lateral entryways (TM5/TM8, TM2/TM11). Several of these events resulted in partial “rocker-switch” conformational changes of the protein from the OFS to the OcS, causing the complete enclosure of the substrate by protein in a headgroup-down conformation (referred to as Substrate Binding Site 1 in the manuscript; Figure 11a) – consistent with a “trap-and-flip” adaptation of the “rocker-switch” mechanism. In our earlier study (Cater *et al.*, *Nature* 2021), we used cryo-EM to reveal that in the IFS, the substrate is also bound in a headgroup-down conformation with its hydrophobic tail entirely enclosed by the protein (referred to as Substrate Binding Site 2 in the manuscript; Figure 11b). We further demonstrated using MD simulations run from this IFS (as this Reviewer suggests we perform) that substrate can be released into the inner leaflet of the membrane through the lateral gateway that is formed between TM5 and TM8. Again, these findings are consistent with a “trap-and-flip” adaptation of the “rocker-switch” mechanism. So, while neither the previous nor the current study elucidate how the fully engulfed substrate diffuses from Site 1 into Site 2, or how the protein transitions from the OcS to IFS, the two studies together strongly suggest that MFSD2A utilizes a “trap-and-flip” adaptation of the “rocker-switch” mechanism for substrate transport rather than a “credit-card” adaptation of the “rocker-switch” model (which would have involved observation of the substrate headgroup sliding through a hydrophilic conduit formed at the protein-membrane interface with its hydrophobic tail remaining embedded in the lipid membrane). We hope that this explanation and the adaptations to the revised manuscript (pages 4-5, 18-20) clarify our conclusions regarding the mechanism.

We appreciate the Reviewer’s very thoughtfully constructed set of computational strategies to capture relevant conformational transitions, especially the OcS to IFS transition that we have not yet recapitulated in molecular detail. As stated above, at the onset of the current study our key questions were: how do lysolipid substrates enter and bind to MFSD2A, and how does this substrate binding instigate conformational changes in the protein? We therefore focused our attention on these important and heretofore unknown aspects of the mechanism. Through our choice of methodologies, we were able to sufficiently detail these aspects of the mechanism and our findings align well with other published results in the field (Cater *et al.*, *Nature* 2021; Wood *et al.*, *Nature* 2021; Martinez-Molledo *et al.*, *NSMB* 2022; Chua *et al.*, *PNAS* 2023). Therefore, while indeed an interesting question, we feel that an in-depth exploration of the OcS to IFS conformational transition is beyond the scope of what the current study aims to achieve, and is perhaps more appropriate for a follow up study so as to give this important question the full effort and focus it deserves.

MINOR COMMENTS

3) It is quite fascinating how the Deep Neural Network (DNN) classification method can succeed with such little data, as the authors used this method on only 4000 frames per ensemble. It would be interesting to see the authors compare the features identified from the DNN method to other feature classification techniques, such as residue-residue contact score (RRCs; <https://elifesciences.org/articles/50279>). The authors deserve to emphasize the success of their DNN technique in classifying critical residues with minimal need for input data.

We thank the Reviewer for their enthusiastic praise. Please refer to our response to Question 1 regarding the DNN method. It is also worth noting, as the Reviewer is likely aware, that accumulating more data doesn't always improve model predictions in machine learning modeling. This depends on various factors, such as the quality of the initial dataset and how easily separable the data sets are for classification.

4) For each of the different ensemble-based results, the independent replicates are of different lengths. Can the authors comment on why the lengths of individual trajectories pooled for cumulative sampling vary? While it is understandable that the lengths of different trajectories can be different for some computing-dependent reasons, but I noticed that different system setups have differing simulation run lengths, which appeared odd. Justifying the reason for why these sets of trajectories are of different length is great for transparency’s sake.

The 48 ggMFSD2A simulation replicates in a POPC/LPC-18:1 membrane were conducted in two batches of 24 replicates each. In one batch, all 24 replicates were simulated for 1.6 microseconds, whereas in the second batch, the 24 replicates were simulated for 1.7 microseconds. As a result, not all replicates in this set had the same duration. This detail is now clarified in the Methods section of the revised manuscript (lines 172-173). In the other simulation sets (ggMFSD2A OcS in a POPC/LPC-18:1 membrane or E312D ggMFSD2A in a POPC/LPC-18:1 bilayer), all replicates were of the same lengths (2.1 and 1.5 microseconds, respectively).

5) I was not able to understand the rationale behind the LPC/POPC membrane setup until I read further into the paper. The authors should include their reasoning behind the dual component bilayer setup in the Methods section as well.

We apologize for the sparse explanation and have now clarified this point in the revised manuscript (lines 162-166).

6) Can the authors confirm that they used the MonteCarloMembraneBarostat in their OpenMM simulations?

We thank the Reviewer for bringing up this inadvertent omission. All of the OpenMM simulations performed for this study did indeed use the MonteCarloMembraneBarostat, as now stated in the revised manuscript (line 156).

7) The language used in the Discussion is very passive, or rather, not deliberate enough. Specifically, I felt this when it comes to describing what mechanism what exact mechanism – “trap-and-flip” or “credit-card” mechanism. The authors state that “they propose a refined model of MFSD2A-mediated transport that combines elements of both the ‘trap-and-flip’ and ‘rocker-switch’ models.” The trap-and-flip model of transport is not at the same level of mechanistic description as rocker-switch; that is, rocker-switch is a fundamental mechanistic description for alternate access, while trap-and-flip is a description of substrate translocation and not conformational change associated with alternate access. Assuming that a reader reads through this paper quickly or is not as familiar with the transporter field terminology, the writing of these mechanistic terms as is misleading because of the different levels of generality (i.e., suggests transport can be either/or). However, there is some nuance here in that perhaps it is possible that the credit-card mechanism does not strictly adhere to the tenets of alternate access? Again, the authors need to better layout their discussion to contextualize their results in light of transporter alternate access schemes. Rather than just restate the results, the authors should firmly state what the translocation strategy is in clear terms of their results once the major comments have been addressed.

Please see our response to Question 2 above and the revisions made to the manuscript (pages 4-5, 18-20) to clarify these important points.

Reviewer #2:

Khelashvili et al has studied the MFSD2A protein and its lipid penetration/binding process using an extensive set of molecular dynamics simulations. While the volume of the data generated is impressive and the methodology is generally sound, I am a bit hesitant to agree with the authors on the significance of their findings.

1. A number of standard analyses are done. Many other types of analyses could be also helpful and may capture important information (rotational motion of helices, pore size dynamics, water profiles, PCA, dynamic network analysis, etc). The MFS family members have been studied extensively and the current manuscript fails to look at many interesting aspects of the structural dynamics of MFSD2A when compared to some other publications. Although there are certainly a number of interesting observations reported in the current manuscript, I think there is more to do with these trajectories.

We thank the Reviewer for these suggestions. In response, we have carried out several new analyses including motions of helices, quantifying allostery in the protein, and water pore analysis (for the pore analysis please see our response to Question 3 below).

Analysis of the helical motions revealed interesting dynamics in TM8 whereby this helix undergoes a kinking motion around a conserved proline in TM8 (P345) during the OFS to OcS transition. It is possible that this kinking may be mechanistically important for regulating the equilibrium between the open/closed states of the TM5/TM8 lateral gateway through which lipids exit the central cavity. This analysis is now presented in the revised manuscript on lines 212-214 (in Methods) and 385-392 (in Results) and in modified Fig. 6d-e).

The analysis of allostery using an information theory based NbIT approach quantitatively demonstrates that the dynamic changes at the intracellular ends of TM7/TM11 in the occluded state are allosterically coupled to fluctuations in the central region of the protein where the substrate headgroup binds. This analysis is now presented in the revised manuscript (see new Fig. 10, lines 224-252 in the Methods, and lines 454-491 in the Results). The inhouse python-based NbIT code used for this analysis is freely available via the following GitHub repository: <https://github.com/weinsteinlab/NbIT>

Overall, we believe that these new results (together with the water pore analysis described below in response to Question 3) have enriched this manuscript and brought to light additional molecular insights into the transport mechanism of MFSD2A. Again, we thank the Reviewer for their suggestions.

2. Although there is no doubt that an extensive set of simulations is performed here, the simulations are still too short and too few to capture longer timescales that may be associated with the entrance and binding of lipids here. For instance, at longer timescales a different pathway and a different binding site may become relevant with a slower kinetics but a favored thermodynamics.

We are grateful to the Reviewer for this comment. We certainly do not dismiss the possibility of other thermodynamically favorable but kinetically slower pathways for substrate binding, and we agree with the Reviewer that such pathways might indeed be revealed through much longer simulations. It is important to note however, that the pathways identified in our study align with two independently obtained experimental results that demonstrate the importance of residues within the TM5/TM8 and TM2/TM11 lateral entryways for MFSD2A-mediated transport (Wood *et al.*, *Nature* 2021, Chua *et al.*, *PNAS* 2023). Moreover, a similar lipid substrate binding mechanism was recently described for the proton-dependent MFS lipid transporter LtaA (Ref. 17 in the manuscript, Lambert *et al.*, *Nat. Comm.*, 2022). We discuss this point in the revised manuscript's Discussion section (lines 521-530).

3. An analysis of the pore size and/or water profile along the pore is missing in this study. I think such analyses would be of interest to the readers. Fig. 6 tries to address these indirectly but much better analysis techniques are already established in the literature that can be used.

We thank the Reviewer for this excellent suggestion. We now present a new analysis of the water pore in the OFS and OcS in the revised manuscript which we performed using the software package HOLE (see new Fig. 7, lines 215-223 in the Methods, and lines 393-402 in the Results). This analysis demonstrated that there is a decrease in the pore radius within the central region (bounded by residues F399 and E312) in the OcS as compared to the OFS, and that the axis of this pore on the EC side of the protein changes during the conformational transition from the OFS to the OcS. We have connected these differences in the water pore to the helical motions we observed during the occlusion process as a part of our analysis. We believe that this offers novel mechanistic insights into the occlusion mechanism which will be of great interest to the readers.

4. The machine learning technique used in this work reveals the importance of TM7 in conformational changes. Some of the other helices are also mentioned in the study. It is useful to compare these findings to some similar findings in the literature with respect to role of TM helices like 1 and 7 or 5 and 11.

Thank you for this comment. We have expanded the Discussion section in the revised manuscript (page 19) to include comparisons between the mechanisms for MFSD2A-mediated transport we propose in this manuscript with that of other MFS transporters. Such comparison reinforces our conclusion that MFSD2A utilizes a

combination of “rocker-switch” (for protein conformational dynamics) and “trap-and-flip” (for the mode of lipidic substrate translocation) mechanisms to mediate transport of this atypical MFS substrate.

5. The authors use ensemble MD to describe a set of simulations consisting of 24-48 replicates. I do not believe this is an appropriate use of the term ensemble. A statistical ensemble consists of a large number of microstates that form one or more macrostates. An ensemble MD would ideally use millions of replicates or perhaps thousands with the current computation standards. In this case, a few dozens of simulations are performed each for ~ 1-2 us. I think what is more appropriate is multi-replicate MD or something of that sort rather than ensemble MD. Now the microstates that are generated may be called an ensemble or a representative ensemble with some abuse of terminology, which is fine. So I am more concerned with the "ensemble MD" terminology that may imply a much larger number of trajectories.

We thank the Reviewer for their insights and as per this suggestion have renamed “ensemble” to “multi-replicate” throughout the manuscript.

Reviewer #3:

In this study, Khelashvili et al use molecular dynamics (MD) to address the mechanism of lysophosphatidylcholine (LPC) transport by MFSD2A at the blood brain barrier. They run MD simulations on a chicken homology model of the outward facing structure to show how the LPC enters the transporter from the outer leaflet. This complements their previous work on the inward facing structure of the chicken protein. They then apply machine-learning based analysis to uncover the allostery between LPC binding and transporter conformational changes. In their analysis, they also highlight interesting effects of the Na⁺ ion on the transport mechanism.

The methodology is sound, the results are convincing, and the manuscript is well written. I found it an interesting read, and think their conclusions are well justified.

We thank this Reviewer for this overall positive evaluation of our work.

I suggest some minor changes which I feel would bolster the study:

1) The simulations use a simple membrane model. This is understandable, but it would be worth mentioning in the discussion that a more complex membrane could potentially impact the observations, especially the presence of lipids such as cholesterol or sphingomyelin.

We thank this Reviewer for raising this important issue. We agree that lipid composition can significantly impact the conformational dynamics of MFSD2A. As we gain more knowledge about the system, exploring this direction is certainly something we plan to do in future studies. We have added a section in the Discussion section (lines 563-566) that addresses this point.

2) I think an image of LPC vs POPC would be useful to guide the readers.

We agree with this suggestion and added a new panel to Fig. 2 (panel g) to incorporate comparison of chemical structures of LPC-18:1 and POPC.

3) In Fig 2d, there is a striking difference between LPC and POPC entry into the transporter, especially via the TM5/TM8 route. Is there a physical explanation for this selectivity? Related to this, does the additional tail or free OH of LPC make any specific interactions with MFSD2A which could explain the selectivity?

Thank you for your comments. From a mechanistic standpoint, we believe that the reason POPC only partially penetrates the transporter, while LPC is able to fully embed, can be explained by a significant difference in the volume of the two lipids. The volumes of POPC and LPC-18:1 at 40°C are estimated to be approximately 1265

Å³ and 822 Å³, respectively (DiPasquale *et al.*, *Methods in Molecular Biology* 2022, Ref. 44 in the manuscript). In fact, in our previous MD simulations of the inward-facing MFSD2A, we observed similar behavior where POPC lipids could only partially and transiently enter the protein cavity on the intracellular side, whereas lysolipid substrates (LPC-18:1, LPC-18:3, LPC-DHA) could extensively penetrate the central cavity (Cater *et al.*, *Nature* 2021; Extended Data Fig. 8). We have added this explanation to the revised manuscript (lines 279-284). We would also like to note that MFSD2A generally shows selectivity for different lysolipids. Experiments reveal that it only transports zwitterionic lysolipids with hydrocarbon tail lengths of 14 or longer. This is intriguing, and our future studies will address the molecular basis for this selectivity.

4) It would sometimes be useful to see the LPC substrate in more detail (i.e., as sticks). This is especially the case in Fig. 3d, to visualise LPC-LPC contacts and the LPC-Na-E312 bridge in detail. This would make a good additional panel, either to the main figures or supplementary data.

Thank you for this excellent suggestion. To improve the visualization, we have modified Fig. 3 to show the LPCs as sticks and also added new panels to this figure, panels e-f, which zoom in to the central region of the protein, highlighting the modes of LPC binding in the two trajectories.

5) Were default protonation states used for all side chains? Were any pKa calculations run to confirm this is the case?

Yes, the default protonation states for all the titratable residues were predicted with pKa calculations at pH 7 using ProPka 3.1. This is now stated in the revised manuscript (lines 140-141).

6) The DNN used here, and previously described by the authors, seems like a very interesting tool. I would encourage them to make the code available if it is not already (I couldn't see it).

Thank you for this positive evaluation of the DNN method. Per suggestion, and for transparency, we made the DNN code freely available via the following GitHub repository: <https://github.com/weinsteinlab/DNN>.

7) I also request the authors make their chicken homology model available via an open access online repository.

A pdb-format coordinate file for the chicken homology model of MFSD2A in the OFS has now been provided as a supplementary file for this submission, and has also been uploaded to the online public repository Zenodo: <https://zenodo.org/record/7749355#.ZBYNPOzMIqw>

*Signed,
Robin Corey*

REVIEWERS' COMMENTS

Reviewer #1 (Remarks to the Author):

Authors have successfully addressed all my comments. I would like to congratulate authors for producing this excellent research work.

Reviewer #2 (Remarks to the Author):

The authors have responded to most of my comments appropriately. However, I am concerned about what has been reported on pore radius. They claim in the response letter that "This analysis demonstrated that there is a decrease in the pore radius within the central region (bounded by residues F399 and E312) in the OcS as compared to the OFS". They also claim the same thing in the revised manuscript (page 14). However, I see the opposite in Fig. 7 (if Fig. 7 is labeled correctly).